# Quantifying hard coal mines CH$_4$ emissions from TROPOMI and IASI observations using the wind-assigned anomaly method

Qiansi Tu[1,2], Matthias Schneider[2], Frank Hase[2], Farahnaz Khosrawi[2], Benjamin Ertl[2,3], Jaroslaw Necki[4], Darko Dubravica[2], Christopher J. Diekmann[2,a], Thomas Blumenstock[2], Dianjun Fang[1,5]

[1]School of Mechanical Engineering, Tongji University, Shanghai, China
[2] Karlsruhe Institute of Technology (KIT), Institute of Meteorology and Climate Research (IMK-ASF), Karlsruhe, Germany
[a] now at: Telespazio Germany GmbH
[3] Karlsruhe Institute of Technology, Steinbuch Centre for Computing (SCC), Karlsruhe, Germany
[4]AGH – University of Science and Technology, Krakow, Poland
[5]Qingdao Sino-German Institute of Intelligent Technologies, Qingdao, China

*Correspondence to*: Qiansi Tu (qiansi.tu@kit.edu), Matthias Schneider (matthias.schneider@kit.edu)

**Abstract.** Intensive coal mining activities in the Upper Silesian Coal Basin (USCB) in southern Poland are resulting in large amounts of methane (CH$_4$) emissions. Annual CH$_4$ emission reached 448 kt according to the European Pollutant Release and Transfer Register (E-PRTR, 2017). As a CH$_4$ emission hot spot in Europe, it is of importance to investigate its emission sources and make accurate emission estimates.

In this study, we use satellite-based total column-averaged dry-air mole fraction of CH$_4$ (XCH$_4$) from the TROPOspheric Monitoring Instrument (TROPOMI) and tropospheric XCH$_4$ (TXCH$_4$) from the Infrared Atmospheric Sounding Interferometer (IASI). In addition, the high-resolution model forecast XCH$_4$ and TXCH$_4$ from the Copernicus Atmosphere Monitoring Service (CAMS) are used to estimate the CH$_4$ emission rate averaged over three years (November 2017 to December 2020) in the USCB region (49.3° - 50.8° N and 18° - 20° E). The wind-assigned anomaly method is first validated using CAMS forecast data (XCH$_4$ and TXCH$_4$), showing a good agreement to the CAMS-GLOB-ANT inventory. It indicates that the wind-assigned method works well. This wind-assigned method is further applied to the TROPOMI XCH$_4$ and TROPOMI+IASI TXCH$_4$ by using the Carbon dioxide and Methane (CoMet) inventory derived for the year 2018. The calculated averaged total CH$_4$ emission over the USCB region is about 496 kt/year (5.9E26 molec./s) for TROPOMI XCH$_4$ and 437 kt/year (5.2E26 molec./s) for TROPOMI+IASI TXCH$_4$. These values are very close to the ones given in the E-PRTR inventory (448 kt/year) and the ones in the CoMet inventory (555 kt/year) and are thus in agreement with these inventories. The similar estimates of XCH$_4$ and TXCH$_4$ also imply that for a strong source, the dynamically induced variations of the CH$_4$ mixing ratio in the upper troposphere and lower stratosphere region are of secondary importance. Uncertainties from different error sources (background removal and noise in the data, vertical wind shear, wind field segmentation, and angle of the emission cone) are approximate 14.8% for TROPOMI XCH$_4$ and 11.4% for TROPOMI+IASI TXCH$_4$. These results suggest that our wind-assigned method is quite robust and might also serve as a simple method to estimate CH$_4$ or CO$_2$ emissions for other regions.

# 1 Introduction

Atmospheric methane ($CH_4$) is the second most important anthropogenic greenhouse gas (GHG) with a larger global warming
potential than carbon dioxide ($CO_2$) (IPCC, 2014). The globally averaged amount of atmospheric $CH_4$ has increased by 260%
to 1877 ± 2 ppb from the preindustrial era until 2019 (World Meteorological Organization, 2020). Methane sources induced
by anthropogenic activities include fossil fuel production and use (e.g., coal mining, gas/oil extraction), waste disposal, and
agriculture, which in total accounts for about 60% of the total $CH_4$ emissions (Saunois et al., 2020). Although most sources
and sinks of $CH_4$ have been characterized, their spatial-temporal variations and relative contributions to the atmospheric $CH_4$
level are still highly uncertain (Kirschke et al., 2013; Saunois et al., 2020).

Approximately 33% of the $CH_4$ emissions from coal mining (42,000 kt/year) are estimated to come from the total fossil-
fuel-related emissions during 2008-2017 (Saunois et al., 2020). $CH_4$ is released primarily to the atmosphere via ventilation
shafts located at the surface during the production and processing of the coal (Saunois et al., 2020; Andersen et al. 2021). The
largest contribution of $CH_4$ emission related with the coal mining activities in Europe is from southern Poland—the Upper
Silesian Coal Basin (USCB) (Luther et al., 2019; Krautwurst et al., 2021). The USCB is in the Silesian Upland, which is a
plateau between 200 and 300 m above sea level with a predominant south-west wind. The USCB within Poland covers an area
of over 5800 km², and to its south is the Tatra Mountain ridge with elevations larger than 2000 m a.s.l. The European Pollutant
Release and Transfer Register (E-PRTR, 2017; https://prtr.eea.europa.eu/, last access: 25 October 2021) reports that the total
$CH_4$ emissions from the USCB region amount to 448 kt/year. Most of these emissions are from mining activities and heavy
industry (Kostinek et al., 2021), which makes this region a hot spot of $CH_4$ emission in Europe.

To investigate the $CH_4$ emission from this hot spot, the Carbon Dioxide and Methane (CoMet) campaign was performed,
covering roughly 3 weeks from May to June 2018. A variety of state-of-art instruments, including in situ and remote sensing
instruments on the ground and aboard five research aircraft, were deployed in order to provide independent observations of
GHG emissions on local to regional scale and provide data for satellite validation (more details can be found in Luther et al.,
2019; Fiehn et al., 2020; Gałkowski et al., 2021; Kostinek et al., 2021; Krautwurst et al., 2021; Wolff et al., 2021). For example,
Gałkowski et al. (2021) present results of in situ GHG measurements obtained over nine research flights of the German research
Aircraft HALO (High Altitude and LOng Range Research Aircraft) acting as the airborne flagship of the CoMet campaign,
together with simultaneous flasks measurements for isotopic composition of $CH_4$. A new lidar CHARM-F ($CO_2$ and $CH_4$
Atmospheric Remote Monitoring Flugzeug) was also onboard HALO and its measurements were investigated to determine
$CO_2$ emission rates from the power plant (Wolff et al., 2021). Many studies present similar $CH_4$ emission estimates for the
region based on different instruments and methods. Luther et al. (2019) estimated $CH_4$ emissions ranging from 6 ± 1 kt/year
for a single shaft to up to 109 ± 33 kt/year for a subregion of the USCB covering several shafts, by using several portable
Fourier Transform Infrared (FTIR) spectrometers (Bruker EM27/SUN). Active AirCore system aboard an unmanned aerial
vehicle (UAV) was used to measure $CH_4$ downwind of a single ventilation shaft and emission rates ranging from 0.5 to 14.5
kt/year based on a mass balance approach and ranging from 1.1 to 9.0 kt/year based on an inverse Gaussian method were

estimated (Andersen et al., 2021). Fiehn et al (2020) analyzed aircraft- and ground-based in situ observations and reported an emission estimate of $436 \pm 115$ kt/year and $477 \pm 101$ kt/year from two selected flights. An advanced model approach was introduced by Kostinek et al. (2021) to investigate two research flights in the morning and afternoon, resulting in estimated $CH_4$ emissions of $451 \pm 77$ kt/year and $423 \pm 79$ kt/year, respectively. Another emission estimate based on the observations from the nadir-looking passive remote sensing Methane Airborne MAPper (MAMAP) instrument accounted for 8.8 kt/year to 78.8 kt/year for a sub-clusters of ventilation shafts (Krautwurst et al., 2021). A recent study (Luther et al., 2022) displays a larger emission rate of 414 – 790 kt/year based on a network of four portable FTS instruments (EM27/SUN) during the CoMet campaign.

Launched in October 2017, the TROPOspheric Monitoring Instrument (TROPOMI) on board the Sentinel-5 Precursor satellite provides an unprecedent high spatial resolution ($5.5 \times 7$ km$^2$) of the $CH_4$ total column-averaged dry-air mole fraction ($XCH_4$) (Veefkind et al., 2012; Lorente et al., 2021). An a posteriori method has been developed by Schneider et al. (2021) to obtain tropospheric $XCH_4$ ($TXCH_4$) by combining observations from TROPOMI and the Infrared Atmospheric Sounding Interferometer (IASI). This synergetic product is not influenced by the changing tropopause height, and it offers improved sensitivity to the tropospheric variations than the total column $XCH_4$ data from either sensor. The improved real-time forecast data with high resolution ($0.1° \times 0.1°$~9 km $\times$ 9km) are produced by the Copernicus Atmosphere Monitoring Service (CAMS) (Agustí-Panareda et al., 2019; Barré et al., 2021). All data sets provide a large spatial coverage and long-term $XCH_4/TXCH_4$ observations, which help to better estimate $CH_4$ emission in the USCB region.

In Sect. 2 we present the data sets and methodology used in this study to derive estimated $CH_4$ emissions. The results and discussions are presented in Sect. 3. We present a novel wind-assigned method introduced by Tu et al., 2022, which is firstly verified by the CAMS model forecasts and then applied to the TROPOMI $XCH_4$ and TROPOMI+IASI $TXCH_4$ data to estimate the $CH_4$ emissions in the USCB region for the time period from November 2017 to December 2020, together with an uncertainty analysis. Finally, the summary and conclusions are given in Sect. 4.

## 2 Data sets and method

There are over 50 active ventilation shafts in the USCB region (49.3° - 50.8° N and 18° - 20° E), Poland, whose emission rates range between 0.17 kt/year and 41.02 kt/year (Gałkowski et al., 2021) (Figure 4b). Most of them are located near Katowice and further west and southwest of Katowice.

### 2.1 CAMS $CH_4$ forecast and emission inventories

The Integrated Forecasting System (IFS, https://www.ecmwf.int/en/publications/ifs-documentation, last access: 27 October, 2021) from the European Centre for Medium-Range Weather Forecasts (ECMWF) is used in the CAMS atmospheric composition analysis and forecasts system to simulate five-day $CO_2$ and $CH_4$ forecasts (Agustí-Panareda et al., 2019, Barré et al., 2021), as well as other chemical species and aerosols (Flemming et al., 2015; Morcrette et al., 2009). This model is also

used in the operational Numerical Weather Prediction (NWP) system, but with additional modules (Agustí-Panareda et al., 2019). The forecast data used in this study is the same suit as the one used in Barré et al. (2021), where the Cycle 45r1 IFS model cycle was implemented. The CAMS GHGs operational dataset includes analysis and forecast data at medium and high resolution with 137 model levels from the surface to 0.01 hPa (Barré et al., 2021). In this study we will focus on using the high-resolution $CH_4$ forecasts, which have a spatial resolution of $0.1° × 0.1°$ and a temporal resolution of 3h, starting from 00:00 UTC. Here we use the daily averaged CAMS forecasts during 9:00 UTC - 18:00 UTC at each resolution grid point. The corresponding standard deviation (STD) is considered as the noise/error.

The anthropogenic methane emissions used in the global CAMS forecasts are from the CAMS global anthropogenic emission inventory (CAMS-GLOB-ANT, Granier et al., 2019; https://permalink.aeris-data.fr/CAMS-GLOB-ANT, last access: 27 October, 2021). The CAMS-GLOB-ANT inventory is based on the emissions provided by the EDGARv4.3.2 inventory for the time period 2000-2012 (Crippa et al., 2018) and linearly extrapolated to 2020 using the trends from the CEDA global inventory for the time period 2011-2014 (Hoesly et al., 2018). The latest version (CAMS-GLOB-ANT v4.2) was released in March 2020, using the same set-up as v4.1 except for adding the emissions in 2020. The anthropogenic sources in the standard v4.2 are divided into 12 sectors and the agriculture sections are split into three sectors, including livestock, soils and waste burning (https://eccad3.sedoo.fr/, last access: 27 October, 2021). The inventory is provided as monthly mean with the same spatial resolution ($0.1° × 0.1°$) as the CAMS forecast data (Granier et al., 2019).

The monthly averages of the CAMS global anthropogenic emissions for different sectors in the study area of USCB are presented in Figure 1. The emissions from the sectors "agriculture soils" and "solvents" are zeros. The $CH_4$ emitted from ships has 19 orders of magnitude, which are much lower than the other sectors. Thus, these three sectors are not shown here. The sources from agriculture livestock ($1.7E25 ± 4.0E25$ molec./s) amount only 4% of the total emissions in this region. The dominant $CH_4$ sources in this region are fugitive sources from energy production and distribution (e.g. fuel use). With a mean value of $7.9E26$ molec./s and a standard deviation of $2.2E25$ molec./s, they account for 82% of the anthropogenic $CH_4$ emissions in the CAMS-GLOB-ANT inventory ($9.7E26$ molec./s in total). This becomes particularly visible in the spatially overlapping distribution within the USCB (see Figure 2). The seasonal emission variations of the fugitive sector are minor and can be ignored. Therefore, we apply the three-year mean of total emissions at grids with significant emissions without considering seasonal variations in the simple cone plume model (see Sect. 2.3). The total emissions amount to $9.7E26$ molec./s over this study area.

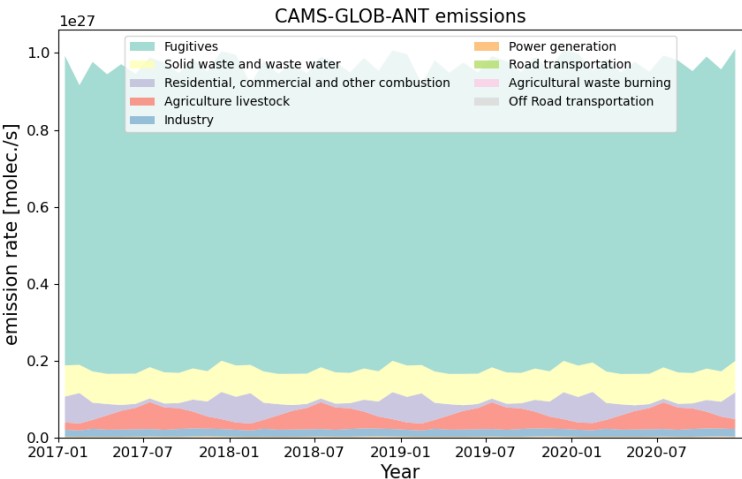

**Figure 1: Stacked area plot for different sectors of the monthly averaged CAMS global anthropogenic emissions (>1E20 molec./s) in the USCB region for 2017-2020 (https://permalink.aeris-data.fr/CAMS-GLOB-ANT, last access: 22 December 2021. Granier et al., 2019).**

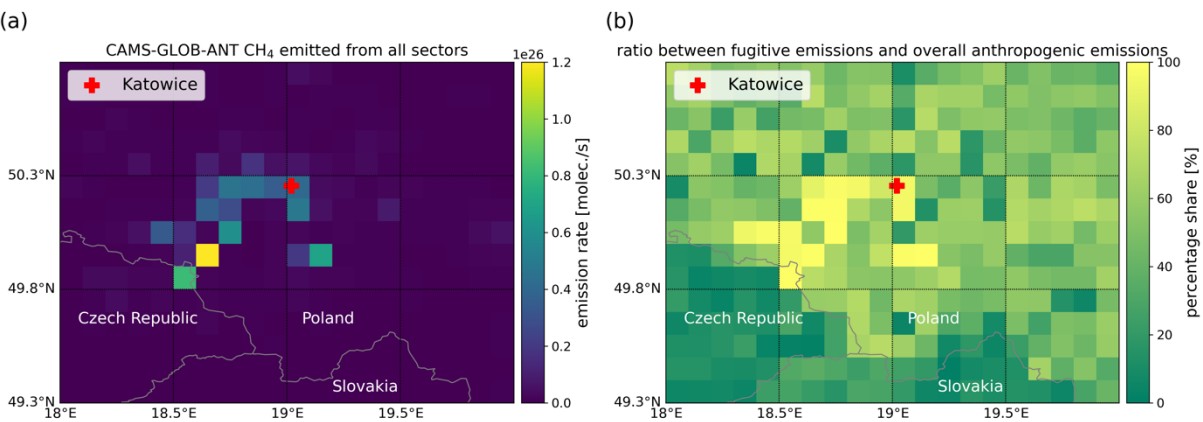

**Figure 2: Spatial distribution of (a) the CAMS global anthropogenic emissions from all sectors and (b) percentage share of the fugitive emissions compared to the overall anthropogenic emissions over the USCB region on a 0.1° × 0.1° latitude/longitude grid. The fugitives are the dominant $CH_4$ sources.**

## 2.2 TROPOMI and IASI data sets

The TROPOMI instrument is a nadir-viewing, imaging spectrometer, which uses passive remote sensing techniques to perform measurements of the solar radiation reflected by and radiated from the earth in the ultraviolet, the visible, the near-infrared and the shortwave infrared spectral bands (Veefkind et al., 2012). The instrument crosses the equator at about 13:30 local solar time at each orbit with a repeat cycle of 17 days. It observes a full swath (2600 km) per second with an orbit duration of 100 min. The algorithm for $CH_4$ column retrieval is called RemoTeC algorithm and it has been extensively used to derive $CO_2$ and

CH4 retrievals from the Greenhouse Gases Observing Satellite (GOSAT) and Orbiting Carbon Observatory-2 (OCO-2; Boesch et al., 2011; Butz et al., 2009, 2011; Hasekamp and Butz, 2008; Schepers et al., 2012). An updated retrieval algorithm has been

implemented by Lorente et al. (2021) to obtain a data suit with less scatter and a higher resolution surface altitude database. This updated TROPOMI XCH4 dataset has been validated with the Total Carbon Column Observing Network (TCCON) (-3.4 ± 5.6 ppb) and GOSAT (-10.3 ± 16.8 ppb), showing very good agreements. In this study the TROPOMI XCH4 during November 2017 and December 2020 within the study area over the USCB region is investigated. The data provided by Lorente et al. (2021) includes an additional quality filter parameter (quality value, qa). TROPOMI XCH4 with qa=1.0 represents the

data under clear-sky and low-cloud atmospheric conditions and the problematic data points are removed as well. This quality filter has been applied in this study and about 16,000 data are derived over the three-year time period considered in this study.

The IASI instrument is a nadir viewing Fourier-transform spectrometer that measures the infrared part of the electromagnetic spectrum. IASI measurements are performed with a horizontal resolution of 12 km and a full swath width of about 2200 km on the ground. It is the key payload element of the polar-orbiting Metop-A -B and -C satellites. These satellites overpass the

150 equator at 09:30 in the morning and 21:30 local time in the evening with about 14 orbits per day. It provides unprecedented accurate vertical information of atmospheric temperature and humidity, which helps to improve numerical weather prediction (NWP) (Collard, 2007; Coopmann et al., 2020). The thermal infrared nadir spectra of IASI have been successfully used in retrieving different atmospheric trace gas profiles and these retrievals are especially sensitive between the middle troposphere and the stratosphere (García et al., 2018; Diekmann et al., 2021; Schneider et al., 2021, 2022). By combining the IASI CH4

profiles and the TROPOMI CH4 total column, which has a higher sensitivity near ground, we are able to detect the tropospheric XCH4 (TXCH4) independently from CH4 at higher altitudes. The combined product cannot be obtained by either the TROPOMI or IASI product independently. The combined product shows a weak positive bias of about 1 % with respect to the reference data (Schneider et al., 2021). We refer to this product in the following as the TROPOMI+IASI TXCH4 and it comprises about 12,000 data points for the time period considered in this study.

**2.3 Simple cone plume model and wind-assigned anomaly method**

The averaged distribution of emitted CH4 over a long-term period can be modeled simply as an evenly-distributed cone-shape dispersion based on the wind and source strength. Since CH4 is a long-lived gas, its decay is negligible for short periods and not considered in the model. This model is referred to as simple cone plume model (see Figure A- 1). This model is easy to apply, and the estimated emission strengths are reasonable compared with the ones from other studies (Tu et al., 2022).

Based on the simple cone plume model, the enhanced CH4 column ($\Delta CH_4$) at the downwind side of the location $(x_i, y_i)$ is computed as:

$$\Delta CH_4(x_i, y_i) = \frac{\varepsilon}{v \cdot d(x_i, y_i) \cdot \alpha} \qquad \text{**Eq. 1**}$$

where the emission strength $\varepsilon$ is the a priori knowledge from the CAMS-GLOB-ANT data set or from the coal mine ventilation shafts in this study (see Sect. 3.2). Their emission rates are assumed to be constant with time from 2017 to 2020. $\alpha$ is the angle

of the emission cone and has an empirical value of 60°, which has been derived from TROPOMI $NO_2$ measurements (Tu et al., 2022). $v$ is the wind speed from ERA5, which is the fifth generation ECMWF reanalysis product using 4D-Var data assimilation and model forecasts in Cycle 41R2 of the ECMWF IFS model (Copernicus Climate Change Service, C3S, 2017, Hersbach et al., 2020). ERA5 provides hourly estimates on 137 pressure levels in the vertical covering the atmosphere from the surface up to 0.01 hPa, with a spatial resolution of 0.25°×0.25° (Hersbach et al., 2020). $d$ is the distance between the downwind location and the $CH_4$ emission source. Each individual source either from the CAMS-GLOB-ANT inventory or from the knowledge of the ventilation shafts is considered as an individual point source. The daily plume from each point source (location at (i,j)) is averaged over daytime (8:00 UTC - 18:00 UTC):

$$\overline{XCH_4}_{(i,j)} = \frac{1}{11}\sum_{t=1}^{11} XCH_{4(i,j),t}$$ **Eq. 2**

these daily plumes are super-positioned over all point sources to obtain a daily plume ($\overline{XCH_4}_{daily}$):

$$\overline{XCH_4}_{daily} = \sum_{s=1}^{N_s} \overline{XCH_4}_{(i,j),s}$$ **Eq. 3**

where $N_s$ represents the number of the sources.

The wind distributions at different height levels (10 m, ~330 m, ~500 m) over the USCB region are presented in Figure 3. The wind speed increases with increasing altitude (see **Table 1**). The ERA5 wind is divided into two opposite wind regimes based on directions (e.g., 135°-315° for SW and the rest for NE). For each wind regime, an averaged plume is computed:

$$\overline{XCH_4}_{SW/NE} = \frac{1}{N_d}\sum_{d=1}^{N_d} \overline{XCH_4}_{daily,d}$$ **Eq. 4**

where $N_d$ is the number of the days with SW wind or NE wind.

The difference of the two plumes is therefore the wind-assigned anomaly:

$$wind-assigned\ anomaly = \overline{XCH_4}_{NE} - \overline{XCH_4}_{SW}$$ **Eq. 5**

The estimated emission strengths can be calculated by fitting the modelled anomalies to the known anomalies from e.g. CAMS $XCH_4/TXCH_4$, and TROPOMI and TROPOMI+IASI observations. Note that $CH_4$ has a lifetime of around 12 years, which results in a high background concentration compared to the newly emitted $CH_4$. Thus, the contributions from the background should be removed for correctly estimating emissions (Liu et al., 2021). The background is considered to consist of a constant value, a linear increase with time, a seasonal cycle, a daily anomaly and a horizontal anomaly. For more details, see the Appendix in Tu et al., 2022. The uncertainties (± values) in the emission estimate are determined by considering the deficits of the background model due to the imperfect elimination of the background and the noise in the data set.

This method was firstly used to estimate the $CH_4$ emission from landfills in Madrid, Spain based on nearly three-year space-borne $XCH_4$ data, and different opening angles were investigated to obtain an empirical value (60°) (Tu et al., 2022). The $CH_4$ emission strengths derived from satellite products have the same orders of magnitude as the ones from single-day observations by ground-based instruments, showing that this method works properly.

**Table 1: Number of days and the averaged wind speed (± standard deviation) per specific wind regime during daytime (08:00 UTC – 18:00 UTC) at different vertical levels from November 2017 to December 2020 over the USCB region. The days for the three-year average coincide with the TROPOMI overpass days.**

| | NE / >315° or <135° | | SW / 135° – 315° | |
|---|---|---|---|---|
| | Number of days in total (%) | Averaged wind speed ± standard deviation (m s$^{-1}$) | Number of days in total (%) | Averaged wind speed ± standard deviation (m s$^{-1}$) |
| **10 m** | 39.1 | 3.2 ± 1.5 | 56. 9 | 3.4 ± 1.6 |
| **~330 m (975 hPa)** | 38.7 | 4.1 ± 2.2 | 56.9 | 4.3 ± 2.3 |
| **~500 m (950 hPa)** | 38.7 | 5.0 ± 2.7 | 57.3 | 5.9 ± 3.5 |

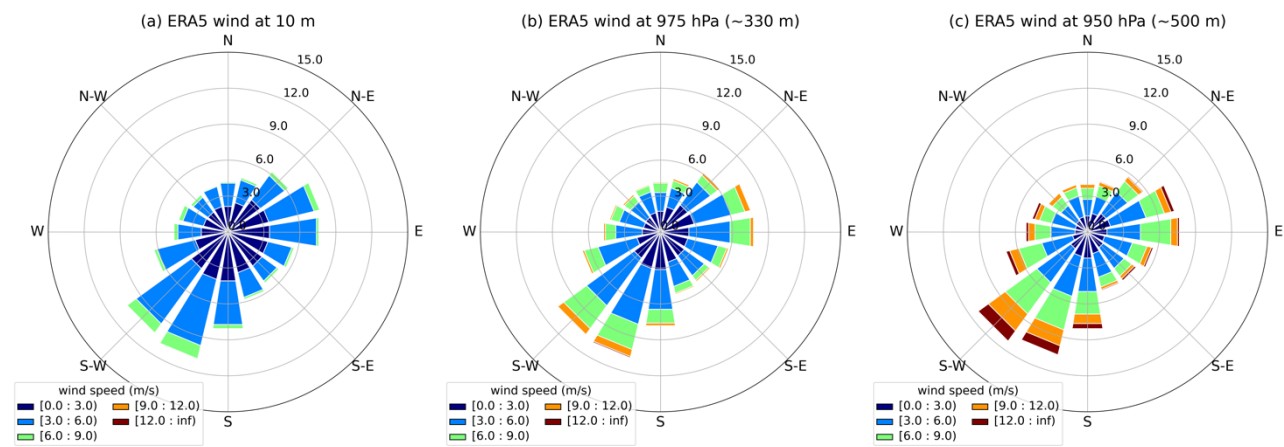

**Figure 3: Windrose plots for daytime (08:00 UTC – 18:00 UTC) from November 2017 to December 2020 for the ERA5 model wind at different vertical levels (10 m, ~330 m and ~500 m). The days for the three-year average coincide with the TROPOMI overpass days.**

## 3. Results and discussion

### 3.1 Estimated emissions derived from CAMS forecasts (evaluation of the method)

The CAMS forecast XCH$_4$ data from November 2017 to December 2020 within the study area are illustrated in Figure 4 left. The areas with high XCH$_4$ amounts fit well with the CAMS anthropogenic CH$_4$ emissions (square symbols). Similar to the CoMet inventory, high sources in the CAMS-GLOB-ANT inventory are centered in this region, but there are other weaker sources outside. The total emission rate of the CoMet inventory is 555 kt/year (6.6E26 molec./s), which is slightly less than the CAMS-GLOB-ANT emissions (815 kt/year). This is probably because the CAMS-GLOB-ANT includes more CH$_4$ emission sources, e.g., wastes, and combustion from residential, commercial, which account to about 20%.

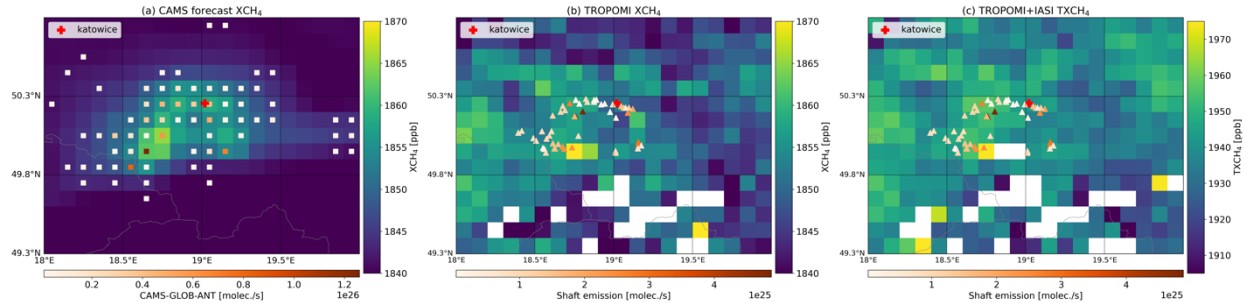

**Figure 4: Averaged (a) CAMS forecast XCH$_4$, (b) TROPOMI XCH$_4$, and (c) TROPOMI+IASI TXCH$_4$ in the USCB region on a 0.1° × 0.1° latitude/longitude grid during November 2017-December 2020. The square and triangle symbols represent the locations of CAMS-GLOB-ANT sources (for better viewing, only the emission strengths larger than 1E24 molec./s are shown here) and the active coal mine shafts from the CoMet inventory (Gałkowski et al., 2021), respectively. Different colors denote the amount of emission rates. The white grids represent no data from TROPOMI or the number of the points in the grid less than 5. A zoom version of panel (b) is shown in the appendix (Figure A- 2). Note, a different colorbar has been used in panel (c).**

Based on the CAMS emissions, the wind-assigned method is applied to CAMS XCH$_4$. The XCH$_4$ enhancement (raw-background) and the wind-assigned anomalies are presented in Figure 5a and b, respectively. The example plumes of the enhancements for wind coming from NE and SW are presented in Figure A- 3. Note, that the CAMS XCH$_4$ is coincided with TROPOMI XCH$_4$ for better comparison. Some data are thus missing here mostly due to the quality filter of TROPOMI observations. After removing the XCH$_4$ background, the XCH$_4$ anomalies well represent the CAMS sources. The highest CH$_4$ sources from the CAMS-GLOB-ANT inventory are also obviously visible in the 2D anomalies. In addition, the spatial distributions of the three XCH$_4$ data products show different patterns (Figure 4), whereas the anomalies (after removing background) patterns are similar (Figure 5(a) and (d), Figure 7(a) and (d)). This indicates that the background removal is of importance for XCH$_4$ and our method works well.

The wind-assigned anomalies for CAMS and simple cone plume model show a very good agreement with a slope of 1.11 and a R$^2$ of 0.85 (Figure 5c). Our results are derived from the CAMS emission information, and they agree very good with the CAMS model data. The estimated emission rate is about 815 ± 17 kt/year (9.7E26 ± 2.0E25 molec./s) when using the ERA5 wind at 975 hPa (~330 m) and this value is quite close to the CAMS-GLOB-ANT (estimated emission rate at other levels are presented in Sec. 3.2, see Figure 8 as well). Therefore, we use ERA5 wind at this level in the following. Note that the points whose distances to the nearest dominant sources are less than 10 km, are removed here, because they are very close to the significant sources and small changes in wind (either speed or direction) can result in high uncertainties.

XCH$_4$ is affected by local surface emissions and a varying stratospheric contribution due to changes in the tropopause altitude (Liu et al, 2021; Schneider et al., 2021). This stratospheric contribution has to be taken into account, in order to be able to use XCH$_4$ for a reliable investigation of local surface CH$_4$ sources and sinks (Pandey et al., 2016). Our background removal method effectively accounts for the stratospheric contribution. To show this we apply the approach to CAMS forecasts of XCH$_4$ (which has a significant stratospheric contribution) and TXCH$_4$ (calculated from the CAMS forecast as the CH$_4$ averaged from surface to 6 km, which should have a very limited stratospheric contribution). The results are presented in Figure 5d-f. The CAMS TXCH$_4$ anomalies have similar distribution as CAMS XCH$_4$ anomalies, suggesting that our

background removal approach reliably removes the stratospheric contribution. The wind-assigned plume and the correlation between CAMS and the wind-assigned model results are very similar between XCH4 and TXCH4. The estimated CH4 emission strength derived from CAMS TXCH4 is 798 ± 15 kt/year (9.5E26 ± 1.8E25 molec./s).

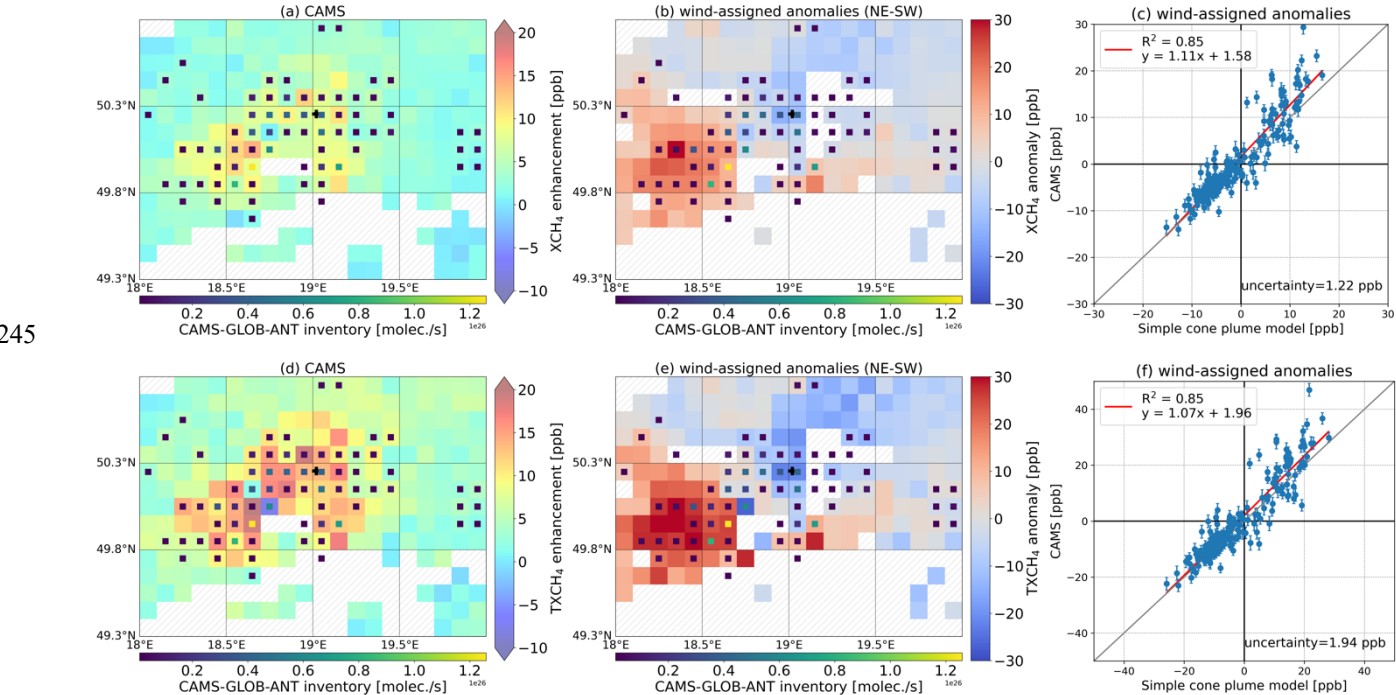


**Figure 5: (a)-(c): CAMS XCH4 enhancement (XCH4-background), the wind-assigned anomalies (NE-SW), and correlation plot of the wind-assigned anomalies between CAMS and the simple cone plume model with using the CAMS-GLOB-ANT inventory (9.7E26 molec./s in total) and ERA5 wind at 330 m during November 2017-December 2020 over the USCB region. (d)-(f): the same as for the upper panel but for CAMS TXCH4. The square symbols represent the locations of the CAMS-GLOB-ANT (>1E24 molec./s) inventory and different colors denote the amount of emission rates. The hatched areas in (a)-(b) and (d)-(e) represent no data in these grids. The uncertainties in (c) and (f) represent the mean error bars, i.e., error propagation of the background uncertainty and the CAMS standard deviation.**


### 3.2 Estimated emissions derived from satellite observations

The high-resolution TROPOMI XCH4 provides the ability to detect and quantify the CH4 emissions (e.g., oil and gas sector, coal mining) on fine and large scales (Pandey et al., 2019; Varon et al., 2019; Gouw et al., 2020; Schneising et al., 2020). Figure 6 illustrates the enhanced XCH4 (raw XCH4-background in the upwind) distribution over the USCB region on an example day (6 June 2018), in which the wind mostly came from northeast. As expected, obvious XCH4 enhancements were observed by TROPOMI along the downwind direction (southwest of Katowice where most ventilation shafts are located), as

well as simulated by the CAMS forecast. The downwind-enhanced XCH4 modeled by our simple cone plume model and based on the CAMS-GLOB-ANT inventory also shows a similar shape of plume. This enhancement was also observed by portable FTIR instruments (COCCON) employed during the CoMet campaign (Figure 4 in Luther et al., 2019). The observations

support the statement that TROPOMI is able to detect the CH₄ emission signals. In addition, the spatial pattern of the downwind plume is similar to that of the cone-shaped plume, which implies our cone-shape assumption is reasonable.

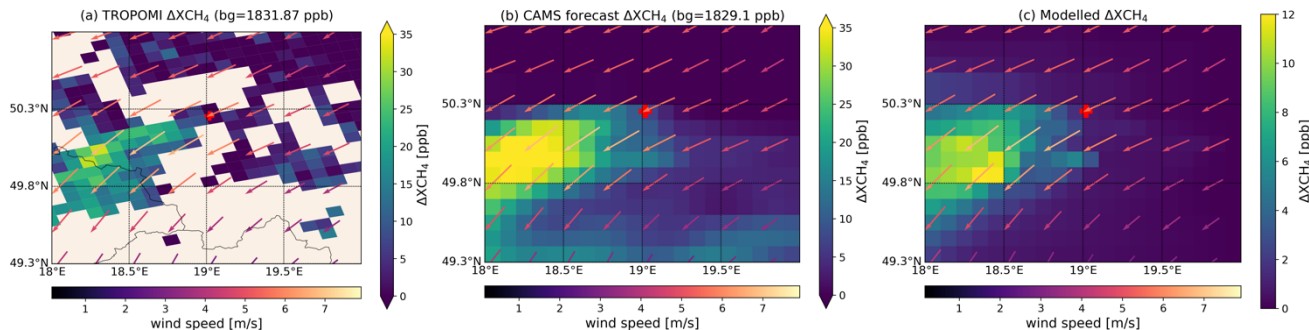


**Figure 6: $\Delta$XCH₄ together with the ERA5 wind at 12:00 UTC from (a): TROPOMI observations at 11:34 UTC, (b): CAMS forecast at 12:00 UTC, and (c): from the simple cone plume model (averaged over the daytime) based on the CAMS-GLOB-ANT inventory over the USCB region on an example day (6 June 2018). The "bg" in the title of (a) and (b) represents the average background, derived from the mean XCH₄ in the upwind region (50.3º-50.8º N, 19.5º -20.0º E). Note, a different colorbar has been used in panel (c) for improved recognizability.**


The three-year averaged TROPOMI XCH₄ observations presented in Figure 4b shows scattered high XCH₄ amounts, whereas CAMS XCH₄ is more concentrated on the center of the study area, and those agree well with its anthropogenic emission sources (CAMS-GLOB-ANT inventory). This might be because TROPOMI detects other real CH₄ sources that are not included in the CAMS forecast model data.

For better comparison with other studies discussing the coal mine emissions in the USCB region, we apply the CoMet inventory as the a priori known sources in the wind-assigned method to estimate the CH₄ emissions. The results are illustrated in Figure 7. The TROPOMI XCH₄ and TROPOMI+IASI TXCH₄ anomalies show high concentrations around the areas where the ventilation shafts are located and the region in the northeast of Katowice. Although the anomalies of the satellite observations are lower than the CAMS results (Figure 5a), their spatial distributions are similar. Positive and negative plumes

can be clearly seen in Figure 7b and e. The correlation of the wind-assigned anomalies between the TROPOMI and simple cone plume model has a very good agreement with an $R^2$ value of 0.76. Similar results are also derived from TROPOMI+IASI TXCH₄ with a $R^2$ value of 0.62. Compared to CAMS data, higher scatter is expected, because satellite observations suffer from observational errors and might contain more CH₄ sources (e.g., landfills, gas distribution network). Although none of these sources have the same orders of magnitude of coal mining emission, they might still bring some errors.

The estimated CH₄ emission strengths are 496 ± 17 kt/year (5.9E26 ± 2.1E25 molec./s) for XCH₄ and 437 ± 27 kt/year (5.2E26 ± 3.2E25 molec./s) for TXCH₄, and both are close to the E-PRTR inventory (448 kt/year). The TROPOMI+IASI result has a higher uncertainty than the TROPOMI result, because (1) the vertical distribution of CH₄ is in general much more difficult to measure than the total column of CH₄ and (2) the vertical distribution is derived by considering two independent measurements, each with its own noise error. This might change for a larger number of data points (e.g., by using data from

more years or by applying the method to IASI and TROPOMI successors on the upcoming METOP-SG satellite, which offers much more collocated observations).

    However, in our study using TXCH$_4$ data in addition to XCH$_4$ data nicely documents the robustness of the method. Important for a correct estimation of the emission is the correct removal of the methane background signal. For XCH$_4$ the stratospheric and the tropospheric backgrounds have to be removed, whereas only the tropospheric background has to be removed for

TXCH$_4$. Despite this difference, we estimate very similar emission rates from both data sets and the emission rate uncertainties of using XCH$_4$ or TXCH$_4$ are small compared to the estimated emission rates.

    Figure 8 summarizes the estimated emission strengths derived from different products based on different a priori knowledge of inventories and wind information at different altitudes (for specific values see Table A- 1). Different a priori inventories result in 16%-32% changes in strength at different altitudes, which is generally smaller than the 47% difference in the total

amount of inventories (9.7E26 for CAMS-GLOB-ANT and 6.6E26 molec./s for CoMet inventory). This is probably due to the different locations of sources and different proportions of each emission source in the total strengths in the two inventories. When using the CAMS-GLOB-ANT inventory, CH$_4$ emission rates derived from CAMS XCH$_4$ and TXCH$_4$ are ~37% and ~56% higher than those derived from TROPOMI XCH$_4$ and IASI+TROPOMI TXCH$_4$, respectively. This difference is mainly due to the difference between the CAMS forecast and satellite products. The strength increases with respect to the increasing

wind speed at higher altitude. Whereas the increment is not always proportional to the wind speed, i.e., less increase in the strength with respect to the wind speed at higher altitude (see Sect. 3.3.1).

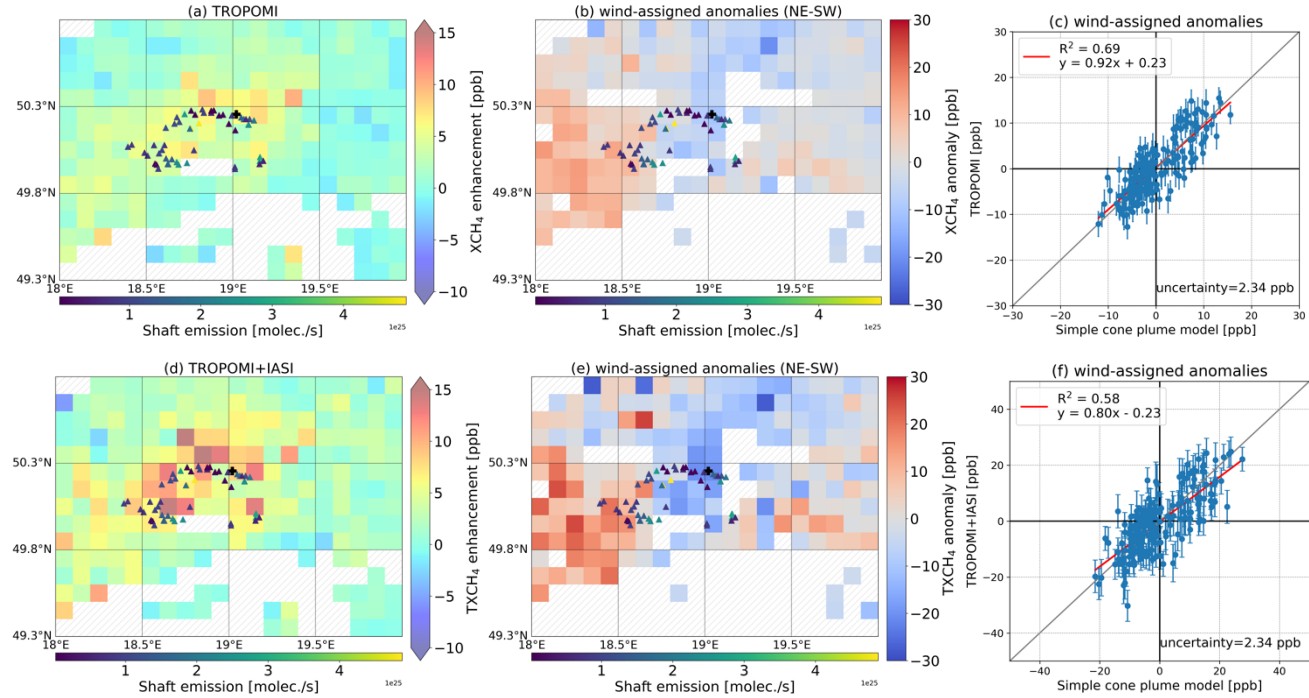

**Figure 7:** Similar to Figure 5, but for (a-c) TROPOMI XCH₄ and (d-f) TROPOMI+IASI TXCH₄. The a priori knowledge of sources are based on the CoMet inventory (6.6E26 molec./s in total, Gałkowski et al., 2021). The triangle symbols represent the locations of the active coal mine shafts and different colors denote the amount of emission rates.

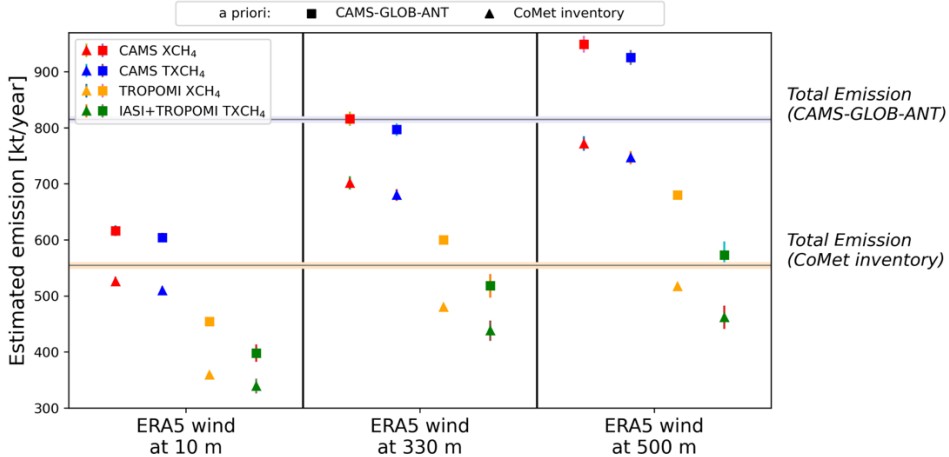

**Figure 8:** Estimated CH₄ emission rates derived from the CAMS forecasts (XCH₄ and TXCH₄), TROPOMI XCH₄, and TROPOMI+IASI TXCH₄ data based on different a priori knowledge of emission sources (CAMS-GLOB-ANT and CoMet inventories) and on ERA5 model winds at different altitudes (10 m, 330 m, 500 m). Square symbols represent the a priori emission sources from the CAMS-GLOB-ANT inventory and triangle symbols represent the a priori emission sources from the CoMet inventory. The two horizontal lines represent the number of total emissions for the CAMS-GLOB-ANT inventory (lavender color) and for the CoMet inventory (orange color), respectively. Note, error bars represent the uncertainties from background removal and noise in the data, which are much smaller than the results and they are not visible here. For specific values see Table A- 1.

## 3.3 Uncertainty analysis

CH₄ signal is weak compared to the background concentration which shows an increasing trend with obvious seasonality and strong day-to-day signals. It is necessary to remove the background signals before estimating the emission strengths. However, the imperfect elimination of the background introduces uncertainties, which can be determined by considering the deficits of the background model and the noise in the background (Tu et al., 2022). In this study, the uncertainties of the estimated strengths include the background uncertainties.

Winds, particularly near the surface, are significantly altered by topography, which yields uncertainties in knowing the transport pathway from emission sources to the measurement location (Chen et al., 2016; Babenhauserheide et al., 2020). Thus, wind is one of the most important factors in correctly estimating the emission rates. Here, we investigate the wind uncertainties based on the CAMS XCH₄ and the CAMS-GLOB-ANT inventory. The wind used in Sect. 3.3.2 and 3.3.3 are from ERA5 at 10 m.

### 3.3.1 Vertical wind shear

Compared to the wind at 330 m, the distributions of wind directions are similar at lower or higher altitudes (10 m and 500 m) but the wind speed increases with higher altitude (Figure 3). The wind speed at 10 m is 20% weaker than that at 330 m (**Table**

1), which yields a corresponding lower emission estimate of 613 ± 13 kt/year (7.3E26 ± 1.5E25 molec./s, -25%) based on the CAMS $XCH_4$ and CAMS emission inventory (Figure A- 4a).

Assuming that the height of the Planetary Boundary Layer (PBL) is typically less than a kilometer, we use the ERA5 wind at 500 m above the ground (Figure 3c) for describing the transport of methane released in the study region. The wind speed at 500 m increases by 22% and 37% for NE and SW regimes, respectively, i.e., 32% on average, compared to the wind at 330 m. The share of SW directed winds is slightly larger at the 500 m level. These differences result in an increase of 13% of the estimated emission rate (924 ± 19 kt/year, 1.1E27 ± 2.3E25 molec./s). The wind speed is linear in the calculation of $\varepsilon$ (Eq. 2), but the wind speeds do not all linearly change for each grid and for each time at different levels. This results in unequal changes between the wind speed and the enhanced columns, and later unequal changes in the estimated emission strength. In addition, the simple cone plume model introduces biases, i.e., the enhanced column in the downwind is set to zero when its location is out of the cone angle (60º). Slight changes in the wind directions might result in a huge difference in the enhanced columns.

### 3.3.2 Use of narrowed angular wind regimes

The long-term wind comes from all directions (0°-360°) (Figure 3). To define the uncertainty of wind regimes' coverage, the wind is separated into two groups with narrow coverage fields: $NE_{1/4}$ (0°-90°) – $SW_{1/4}$ (180°-270°) and $NW_{1/4}$ (270°-360°) – $SE_{1/4}$ (90°-180°). The final estimated emission strength is weighted by the number of days on which, on average, the wind blew in the respective wind regime (i.e., 115 days for $NE_{1/4}$ – $SW_{1/4}$ and 71 days for $NW_{1/4}$ – $SE_{1/4}$, respectively). The $XCH_4$ anomalies and the plume for the $NE_{1/4}$ – $SW_{1/4}$ regime are quite similar to those with using wider-coverage NE and SW fields (Figure 9a-c). The wind-assigned anomalies derived from CAMS and the simple cone plume model show very good agreement as well. Slightly less data points are found here because of the choice of narrower wind fields, especially for $NW_{1/4}$ – $SE_{1/4}$ wind fields. The estimated emission rate is about 773 ± 19 kt/year (9.2E26 ± 2.3E25 molec./s) for the $NE_{1/4}$ – $SW_{1/4}$ field. This indicates that the effect of the segment in the wind field coverage is negligible when there are enough measurements. The use of $NW_{1/4}$ – $SE_{1/4}$ wind fields yields an emission strength of 1176 ± 134 kt/year (1.4E27 ± 1.6E26 molec./s). The higher uncertainty is probably due to less measurements in these wind fields. The weighted rate is therefore about 927 kt/year (1.1E27 molec./s), 13.4% higher than based on the wider NE-SW wind regime (Sec. 3.1).

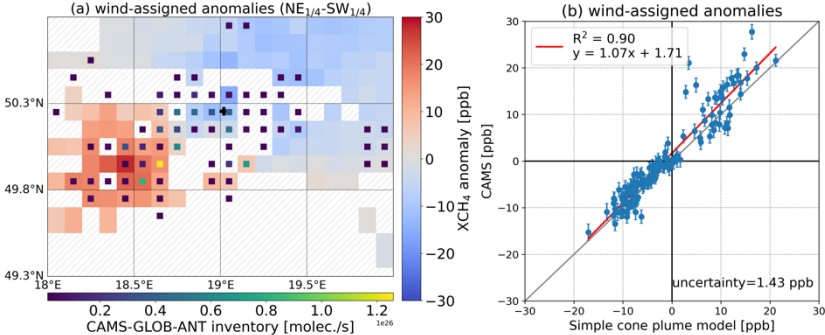

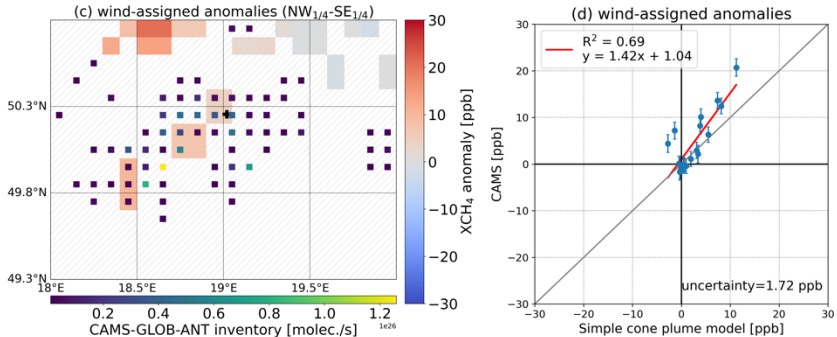

**Figure 9: Similar figures to Figure 5b-c. Results are derived from CAMS XCH$_4$, CAMS emission inventory and ERA5 wind at 330 m for (a)-(b) narrow wind coverage (NE$_{1/4}$ and SW$_{1/4}$), and (c)-(d) narrow wind coverage (NW$_{1/4}$ and SE$_{1/4}$).**

### 3.3.3 Investigation of different choices for wind field segmentation

The wind category here is based on the predominant wind fields over the USCB region and is divided into two opposite regimes (SW and NE). To investigate the effect of the segmentation on the uncertainty in the emission rate estimation, we additionally

apply another kind of segmentation: N (<90° or >270°) and S (90° - 270°) categories. Similar results are found and are shown in Figure 10. Though the 2D distribution of the plume changes due to the new wind category, an obvious plume can be seen. The estimated emission rate is 773 ± 19 kt/year (9.2E26 ± 2.3E25 molec./s), which is only 5.2% less than that using NE and SW wind categories. The correlation of the wind-assigned anomalies derived from the CAMS and the simple cone plume model shows a very good agreement as well, with a similar $R^2$ value of 0.85 to that in the NE-SW wind category. This result

demonstrates that our method is not significantly influenced by the wind regime division.

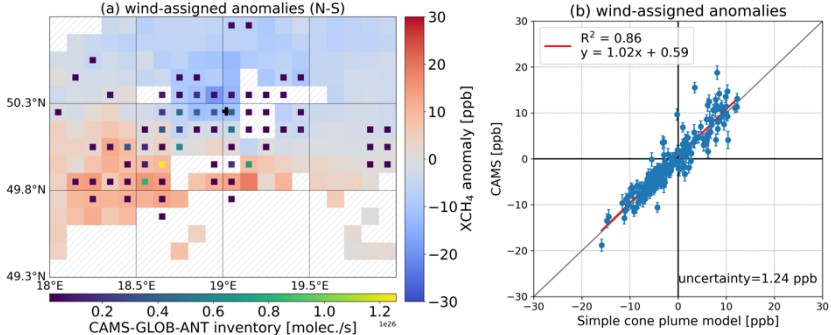

**Figure 10: Similar figures to Figure 5b-c. Results are derived from CAMS XCH$_4$, CAMS emission inventory and ERA5 wind at 330 m but using a new wind category (N and S).**

3.3.4 Investigation of different choices for angle of the emission cone

The angle ($\alpha = 60°$) used in the simple cone plume model is an empirical value which affects the deduced emission strengths. Figure 11 shows the results when $\alpha$ is decreased or increased by 10°. Changes in the spatial distributions of wind-assigned anomalies and in the correlations derived from CAMS and the simple cone plume model are nearly negligible when using different angles in the model. The estimated emissions are 789 ± 16 kt/a (9.5E26 ± 1.9E25 molec./s) for $\alpha = 50°$ and 832 ±

17 kt/a (9.9E26 ± 2.0E25 molec./s) for $\alpha = 70°$, which are 3% lower and 2% higher than that with using the empirical angle
($\alpha = 60°$).

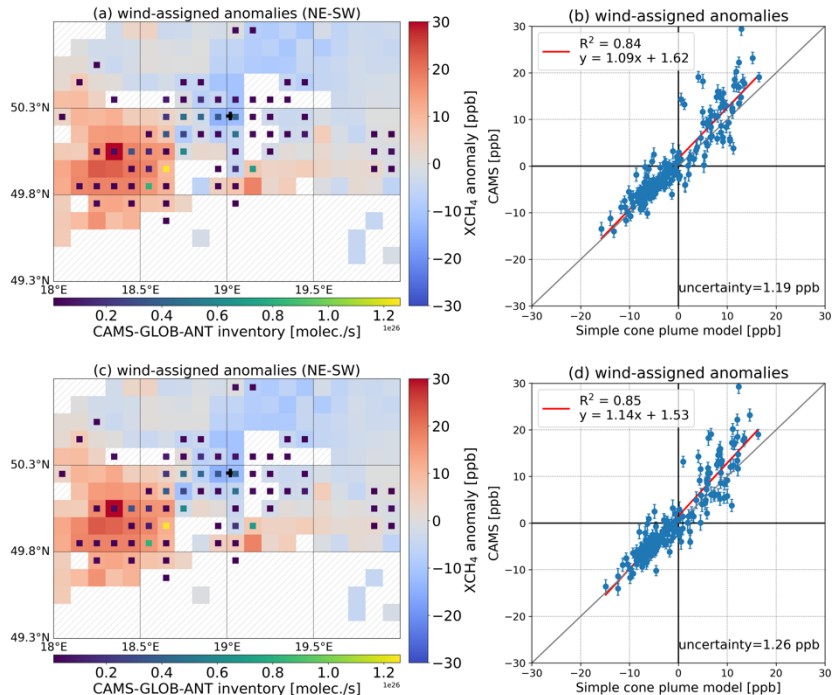

**Figure 11: Similar figures to Figure 5b-c. Results are derived from CAMS XCH$_4$, CAMS emission inventory and ERA5 wind at 330 m for (a)-(b) $\alpha = 50°$, and (c)-(d) $\alpha = 70°$.**

The changes in the estimated emission rates for different products due to different error sources are summarized in Table A- 2. Based on the error propagation, the total uncertainty in the estimated emission rates from the different error sources (background removal and noise in the data, vertical wind shear at 500 m, wind field segmentation, and opening angle $\alpha = 70°$) is approximately 14.7% for CAMS XCH$_4$, 14.8% for TROPOMI XCH$_4$ and 11.4% for TROPOMI+IASI TXCH$_4$. Note that, the use of narrowed angular wind regimes is not a preferable way due to few amounts of data in narrowed wind regimes
and thus, is not considered an error source. In addition, the 500 m wind shear was used as a contribution to the budget, as the 10 m wind is not expected to be representative of the PBL.

## 4. Conclusion

Intensive mining activities are the dominant CH$_4$ emission sources in the USCB region, Poland, where one of the largest coal mining areas in Europe is located. It is thus of importance to quantify the CH$_4$ emissions from this area. In this study we use
the combination of a simple cone plume model and a novel wind-assigned model to estimate CH$_4$ emission rates from high-resolution CAMS forecast XCH$_4$ and TXCH$_4$, along with satellite data (TROPOMI XCH$_4$ and TROPOMI+IASI TXCH$_4$) over the USCB region (49.3°N-50.8°N and 18°E-20°E) from November 2017 to December 2020.

Based on the CAMS-GLOB-ANT inventory, the dominant $CH_4$ source is emitted from energy production and distribution, and the significant sources are spread around the city of Katowice and its southwest region. We firstly apply the wind-assigned method to the CAMS forecasts based on the a priori knowledge of CAMS-GLOB-ANT inventory (815 kt/year, 9.7E26 molec./s in total) and ERA5 wind at ~330 m. We use the wind-assigned anomalies of $XCH_4$/$TXCH_4$ to represent the difference of $XCH_4$/$TXCH_4$ between the conditions of two opposite wind fields (NE and SW). The wind-assigned anomalies derived from CAMS $XCH_4$/$TXCH_4$ show very good agreements with the output of the simple cone plume model with an $R^2$ value of 0.85 for CAMS $XCH_4$ and CAMS $TXCH_4$. This nice correlation indicates that our background removal works well. In addition, similar estimates are derived from CAMS $XCH_4$ (815 kt/year, i.e., 9.7E26 E25 molec./s) and $TXCH_4$ (798 kt/year, i.e., 9.5E26 molec./s).

To investigate the $CH_4$ emissions from this hot spot, the CoMet campaign was performed in 2018. Locations and emission rates of the ventilation shafts of the coal mine used in this study are based on this campaign. Based on this knowledge, the emissions are estimated as 496 kt/year (5.9E26 molec./s) and 437 kt/year (5.2E26 molec./s) from the TROPOMI $XCH_4$ and combined TROPOMI+IASI $TXCH_4$, respectively. These results are 40% less than that derived from the CAMS model and CAMS-GLOB-ANT inventory. It is probably because the CAMS-GLOB-ANT includes many sectors of anthropogenic sources, like wastes, and combustion from residential and commercial, which account for about 20%. Nevertheless, our results derived from satellite observations are close to the E-PRTR inventory of 448 kt/year and reasonably compared to the CoMet inventory (555 kt/year), and to previous studies over the USCB region (ranging from 9 kt/year to 79 kt/year for a sub-cluster of shafts (Krautwurst et al., 2021) up to 477 kt/year derived from one flight (Fiehn et al. (2021)).

Similar 2D anomalies and plumes are also observed for TROPOMI $XCH_4$ and TROPOMI+IASI $TXCH_4$. This nicely documents the robustness of the method. The TROPOMI+IASI result has a higher uncertainty than the TROPOMI result, because (1) the vertical distribution of $CH_4$ is in general much more difficult to measure than the total column of $CH_4$ and (2) the vertical distribution is derived by considering two independent measurements, each with its own noise error. This might change for a larger number of data points (e.g., by using data from more years or by applying the method to IASI and TROPOMI successors on the upcoming METOP-SG satellite, which offers much more collocated observations). Nonetheless, the uncertainties are insignificant compared to the estimated emission rates.

Wind contains uncertainties in knowing the transport pathway from emission sources to the measurement location and thus, we analyze the effects in selecting wind at lower and higher altitudes (10 m and 500 m), wind field coverage, and wind category. Wind distributions at higher levels are similar to the ones at 330 m. However, their speeds decrease by 20% at 10 m and increase by 32% at 500 m, which results in changes in the emission rates by -25% and 13 % for CAMS $XCH_4$, respectively. Narrower wind field coverage (0°-90° for NE regime and 180°-270° for SW regime) and different wind segmentation (<90° or >270° for N regime and 90°-270° for S regime) introduce uncertainties of +13.4% and -5.2% for CAMS $XCH_4$, respectively. The agreements for these sensitivity tests of the wind-assigned anomalies derived from the CAMS and from the simple cone plume model are as good as that using previous NE and SW wind fields. The impact of a suboptimal choice for the angle (60°) used in the simple cone plume model is also discussed. The estimation is decreased by 3% for $\alpha = 50°$ and increased by 2%

for $\alpha = 70°$ for CAMS XCH$_4$. This small change supports the empirical choice for $\alpha$. Based on the error propagation, the total uncertainty in the estimated emission rates from the different error sources (background removal and noise in the data, vertical wind shear at 500 m, wind field segmentation, and opening angle $\alpha = 70°$) is approximately 14.7% for CAMS XCH$_4$, 14.8% for TROPOMI XCH$_4$ and 11.4% for TROPOMI+IASI TXCH$_4$. These results suggest that the wind-assigned method is robust and is also suitable for estimating CH$_4$ and CO$_2$ emissions in other regions.

*Data availability*. The data are accessible by contacting the corresponding author (qiansi.tu@kit.edu). The SRON S5P-RemoTeC scientific TROPOMI CH$_4$ dataset from this study is available for download at https://doi.org/10.5281/zenodo.4447228 (Lorente et al., 2021, last access: 08 November 2021). The TROPOMI data set is publicly available from https://scihub.copernicus.eu/ (last access: 08 November 2021; ESA, 2020). The access and use of any Copernicus Sentinel data available through the Copernicus Open Access Hub are governed by the legal notice on the use of Copernicus Sentinel Data and Service Information, which is given here: https://sentinels.copernicus.eu/documents/247904/690755/Sentinel_Data_Legal_Notice (last access: 08 November 2021; European Commission, 2020). The MUSICA IASI data set is available for download via https://doi.org/10.35097/408 (Schneider et al. 2021).

*Author contributions*. Qiansi Tu, Frank Hase and Matthias Schneider developed the research question. Qiansi Tu wrote the manuscript and performed the data analysis with input from Frank Hase, Matthias Schneider and Farahnaz Khosrawi. Matthias Schneider, Benjamin Ertl and Christopher J. Diekmann provided the combined (MUSICA IASI + TROPOMI) data and supported technically for the analysis of these data. Jarosław Necki supported in consultation of the local situation and CoMet inventory. All authors discussed the results and contributed to the final manuscript.

*Competing interests*. The authors declare that they have no conflict of interest.

*Acknowledgements*. The CAMS results were generated using Copernicus Atmosphere Monitoring Service (2017–2020) information. Neither the European Commission nor ECMWF is responsible for any use that may be made of the Copernicus information or data it contains. We also thank Michela Giusti and Kevin Marsh in the Data Support Team at ECMWF for retrieving and providing comments about the CAMS data. This research has largely benefit from funds of the Deutsche Forschungsgemeinschaft (provided for the two projects MOTIV and TEDDY with IDs/290612604 and 416767181, respectively). Important part of this work was performed on the supercomputer ForHLR funded by the Ministry of Science, Research and the Arts Baden-Württemberg and by the German Federal Ministry of Education and Research. We acknowledge Emissions of atmospheric Compounds and Compilation of Ancillary Data (ECCAD) for providing CAMS-GLOB-ANT inventory data. We also give thanks to Claire Granier from Laboratoire d'Aerologie, Toulouse, France for providing information about the uncertainties of the CAMS-GLOB-ANT inventory.

We acknowledge the support by the Deutsche Forschungsgemeinschaft and the Open Access Publishing Fund of the Karlsruhe Institute of Technology.

## Appendix

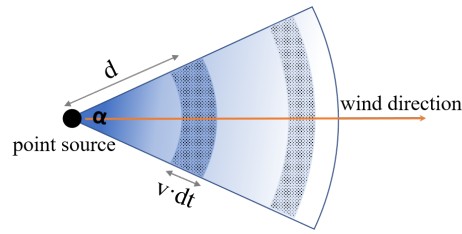

**Figure A- 1: Sketch of the simple cone plume model used to explain the CH₄ emission estimation method. The methane at the point source is distributed along the wind direction (wind speed: $v$) in the cone-shaped area with an opening angle of α. The point source emits the methane at an emission rate of ε. We assumed the methane molecules are evenly distributed in the dotted area A, and the distance from area A to the point source is d. Therefore, the emitted methane in dt time period equals to the amount of methane in the area A. It yields the equation $\varepsilon \times dt \approx \Delta column \times \frac{\alpha}{\pi} \times \pi \times d \times v \times dt$. This figure is adopted from Tu et al. (2022).**

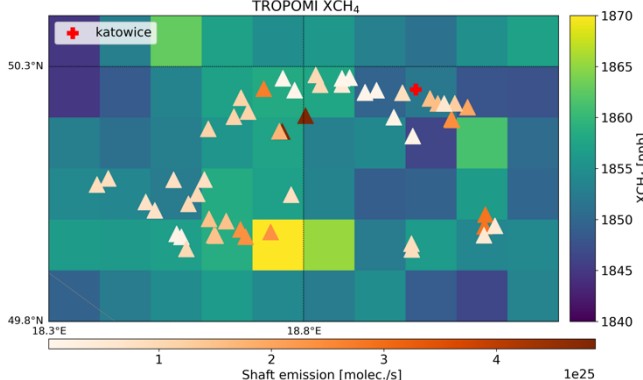

**Figure A- 2: A zoomed figure of Figure 4(b).**

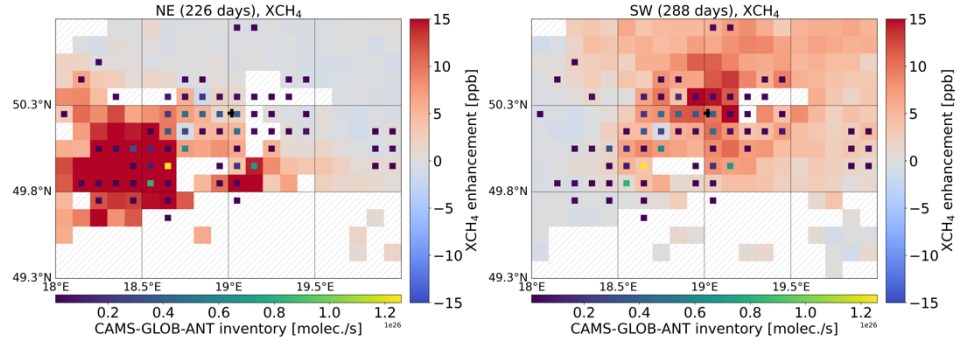

**Figure A- 3: The enhancements for wind coming from NE and SW.**

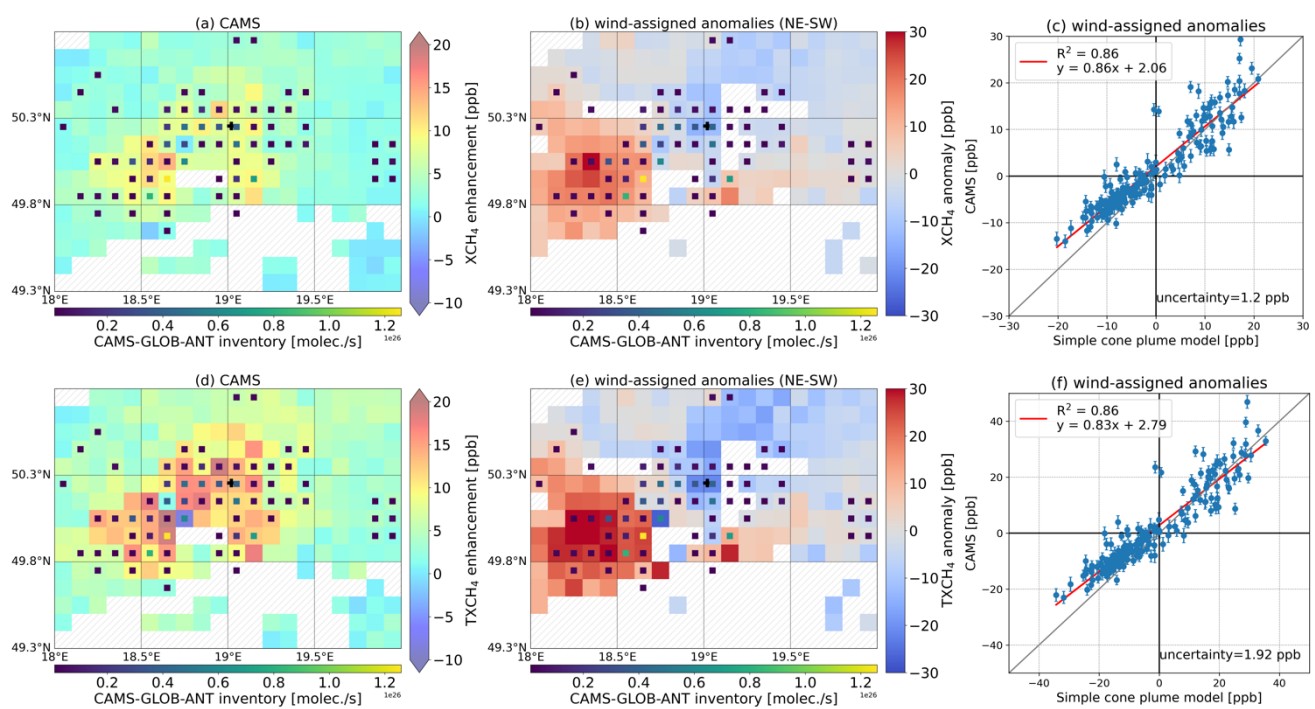

**Figure A- 4: Similar to Figure 5 but using ERA5 wind at 10 m.**

**Table A- 1: Estimated CH$_4$ emission rates derived from CAMS forecasts (XCH$_4$ and TXCH$_4$), TROPOMI XCH$_4$, and IASI&TROPOMI TXCH$_4$ data based on different a priori knowledge of emission sources (CAMS-GLOB-ANT and CoMet campaign inventories) and ERA5 model winds at different altitudes (10 m, 100 m and ~500 m).**

| prior emission sources | ERA5 wind at 10 m | | ERA5 wind at 330 m (975 hPa) | | ERA5 wind at 500 m (950 hPa) | |
|---|---|---|---|---|---|---|
| | CAMS-GLOB-ANT (total = 9.7E26 molec./s) | CoMet inventory (total = 6.6E26 molec./s) | CAMS-GLOB-ANT (total = 9.7E26 molec./s) | CoMet inventory (total = 6.6E26 molec./s) | CAMS-GLOB-ANT (total = 9.7E26 molec./s) | CoMet inventory (total = 6.6E26 molec./s) |
| CAMS XCH$_4$ | 7.3E26 ± 1.5E25 | 6.3E26 ± 1.3E25 | 9.7E26 ± 2.0E25 | 8.3E26 ± 1.8E25 | 1.1E27 ± 2.3E25 | 9.2E26 ± 2.1E25 |
| CAMS TXCH$_4$ | 7.2E26 ± 1.3E25 | 6.1E26 ± 1.2E25 | 9.5E26 ± 1.8E25 | 8.1E26 ± 1.6E25 | 1.0E27 ± 2.1E25 | 8.9E26 ± 1.9E25 |
| TROPOMI XCH$_4$ | 5.4E26 ± 1.8E25 | 4.5E26 ± 1.5E25 | 7.1E26 ± 2.4E25 | 5.9E26 ± 2.1E25 | 8.1E26 ± 2.8E25 | 6.3E26 ± 2.3E25 |

| IASI&TROPOMI TXCH4 | 4.7E26 ± 2.7E25 | 4.0E26 ± 2.3E25 | 6.2E24 ± 3.7E25 | 5.2E26 ± 3.2E25 | 6.8E24 ± 4.3E25 | 5.5E26 ± 3.6E25 |

Table A- 2: Changes in the estimated emission rates for different products when using different input data or under different situations compared to their results using the default setting (wind at 330 m, NE-SW wind segmentation, $\alpha$ =60º, CAMS-GLOB-ANT for CAMS XCH4 and CoMet inventory for TROPOMI XCH4 and TROPOMI+IASI TXCH4).

|  | CAMS XCH4 | TROPOMI XCH4 | TROPOMI+IASI TXCH4 |
|---|---|---|---|
| Background removal & noise in the data | 2.1% | 3.6% | 6.1% |
| Vertical wind shear (500 m) | 13.4% | 6.8% | 5.8% |
| wind field segmentation (N-S) | -5.2% | 12.7% | 7.7% |
| angle of the emission cone ($\alpha=70º$) | 2.1% | 0.07% | -0.02% |
| **Total:** | **14.7%** | **14.8%** | **11.4%** |

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
