# Peer review of "Quantifying hard coal mines CH4 emissions from TROPOMI and IASI observations using the wind-assigned anomaly method"

_Atmospheric Chemistry and Physics, 2022_

## Referee Comment (RC2)

**General comments:**

The manuscript „Quantifying hard coal mines CH4 emissions from TROPOMI and IASI observations using high-resolution CAMS forecast data and the wind-assigned anomaly method" by Qiansi Tu et al., reports on a top-down approach to estimate methane emissions on the region scale. In their work the authors focus on the CH4 emissions from hard coal mines in the Upper Silesian Coal Basin. Their emission estimation is based on applying a simple cone-plume-model and fitting the associated wind-assigned anomalies to enhancements in the XCH4 data retrieved by satellite observations form TROPOMI and IASI over a period of three years. Simple, straight forward to apply approaches, as presented by the authors, to estimate CH4 emissions on a local scale are highly relevant, especially in the light of recent COP 26 and the Global Methane Pledge that emerged from it.

The manuscript is well structured, but poorly written. The high amount of technical errors suggests that the authors have made an insufficient effort in proofreading, before submitting their manuscript to the journal. Nonetheless, I recommend the study as suitable for publication in ACP after the major revision has been addressed.

**Specific comments:**

1. molec/s is a rather small-scaled unit for an observation-period of three years over a 100x100km region. Maybe kt/year is more suitable. Would also get rid of hard to read exponential nomenclature. Furthermore, in the introduction your use of units switches from kt/yr to TgCH4yr-1.

2. Sometimes it is not clear whether the CAMS GHG dataset or the CAMS emission inventory is being referred to (e.g. Lines 188 and 206; title of Fig. 5; caption Fig. 9; …). This can be particular challenging, when the authors' emission estimates retrieved from the CAMS GHG dataset are compared to the CAMS emission inventory. I recommend using "CAMS-GLOB-ANT" throughout the text whenever referring to the emission inventory.

3. Line 21: "wind directions" Throughout the text, the division into wind regimes is designated differently. I think the designations wind regimes/segments/divisions are suitable, wind sectors/sections are not. Please adjust the text accordingly.

4. Line 154: "This model is referred to as simple plume model". A lot of plume models can be described as "simple". I suggest referring to it as "cone plume model". This would have the advantage that the designation is self-descriptive. If the authors want to stick with the term "simple plume model" for consistency with the earlier publication, that's fine with me.

5. Line 162: To assume constant emission rates for three years is rather bold. According to the E-PERTR variations of a few percent are to be expected. Since the uncertainties in your estimate of emission rates are rather small, much of this might be due to interannual fluctuations. Please reconsider this statement and consider estimating emission rates for individual years.

6. In Eq. 2 you introduced $\Delta XCH_4$ as the enhanced $CH_4$ column. Later, especially in figures, you use $\Delta XCH_4$ for the wind-assigned anomalies. Please introduce a distinctive notation for the wind-assigned anomalies to avoid confusion with the enhanced $CH_4$. See also the next comment for that.

7. Line 163: Where does the empirical value of 60° come from? From your earlier publication I know that it comes from TROPOMI $NO_2$ measurements, but this should be explained and cited here.

8. Line 169-170: „For each wind sector, an averaged plume is computed and the difference of the two plumes are therefore the wind-assigned anomalies". Above it is said that daily averaged plumes are calculated. Here, "averaged plume" refers to a plume averaged over all daily-plumes, which propagate NE/SW on daily-average. That means at that step you have two plumes for the entire three years. At each pixel your wind-assigned anomaly is calculated by $\overline{XCH_{4_{SW}}}(i,j) - \overline{XCH_{4_{NE}}}(i,j)$. Without checking your previous publication, I couldn't understand this. A few more equations would be helpful to explain your approach. Something like: $\overline{XCH_{4_{SW/NE}}}(i,j) = \frac{1}{N_d}\sum_d^{N_d} XCH_{4,d}(i,j)$ with $N_d$ = number of days and i,j ∈ SW/NE and wind-assigned anomaly = $\overline{XCH_{4_{SW}}}(i,j) - \overline{XCH_{4_{NE}}}(i,j)$.
   Please consider showing table A-1 here instead of the appendix.

9. Line 175: Background removal is critical for correctly estimating the emission rate. From your earlier publication I know that the uncertainty in the background subtraction is included in the uncertainty of your enhanced XCH4 values. Please add a short statement about uncertainty in background subtraction in chapter 3.3. Moreover, I would be interested in how the background differs between CAMS and TROPOMI data.

10. Line 195: "The XCH4 anomalies (raw-background) and the wind-assigned anomalies are presented in Figure 5a and b, respectively". Please change to XCH4 enhancement, to avoid risk of confusion to the wind-assigned anomalies. This would be more consistent as the term "enhanced" has already been used for background-free XCH4 in the context of Eq.2.

11. Line 196-197: I don't understand what you mean by: „Note, that the CAMS XCH4 is coincided with TROPOMI XCH4 for better comparison". Were the CAMS data interpolated to the TROPOMI grid and accordingly the filtered TROPOMI grids are also missing for CAMS? But why are different grids missing for all plots displaying wind-assigned anomalies?

12. Line 213 ff: "To remove this influence, we calculate the tropospheric CAMS forecasts CH4 (TXCH4) from the surface up to 7 km." Why 7 km? The height of the tropopause surely changes over the course of the three-year observation period.

13. Line 233-234: „In addition, the downwind plume is similar to the cone shaped plume in our simple plume model …" What do you mean by similar? Just the spatial occurrence? As you use three different colormaps it is hard to judge by eye. It is clear, that the modelled cone-plumes result in XCH4 enhancements which are smaller by a factor of two or even more, suggesting that the CAMS-GLOB-ANT emissions are too low.

14. Line 234: "… which implies our model assumption is reasonable." Either CAMS-GLOB-ANT has too small emissions, or the model generates a systematic bias. See comment above.

15. Figure 5 correlation plots (c & f): I assume the gray line is the bisector. Please include your regression line. Please do so also for the other correlation plots in the manuscript.

16. Figure 6: colormaps: Why a diverging colormap for wind speed? Is 4 m/s a representative mean value? If so, please indicate this in the caption, otherwise I would suggest a perceptually uniform sequential colormap. Also, please do not use the same colormaps for windspeed as for XCH4 enhancements or wind-assigned anomalies.

17. Figure 6 colormaps: Why is TROPOMI transparent/shaded and the other two are not? Please have consistent colormaps for all plots. Especially the modeled plume has much smaller values than CAMS and TROPOMI. This becomes more difficult to see with the currently used colormaps.

18. Figure 6: Do I understand correctly that CAMS forecast is at 12 UTC? Is TROPOMI also at 12 UTC? Is the modeled plume an average over 2018-06-06 or also at 12 UTC? Please clarify.

19. Figure 6: As you are showing snapshots of a specific time you should also give the respective value of the background term that has been subtracted. Either in the caption or the title.

20. Figure 5 & 7: Why are the wind-assigned anomalies far more south-west than the CAMS forecasts/observations? Is the ERA5 wind wrong or is there a methodological error?

21. Figure 5 & 7: I assume "CAMS emission (9.7E26 molec./s)" is the sum of all sources in the CAMS-GLOB-ANT inventory and used as an a priori value for the calculation of the wind-assigned anomalies, is this correct? Please clarify in the caption and do not repeat as a title for every single plot. Same for Figure 7.

22. Line 253: "The TROPOMI+IASI result has a slightly higher uncertainty than the TROPOMI result". Please remove "slightly". The uncertainty is more than a factor 4 higher.

23. Line 263: „... the emission rate uncertainties of using XCH4 or TXCH4 are insignificant compared to the estimated emission rates." Please change "insignificant" to "small" or something equivalent. If uncertainties were insignificant, they would not need to be reported.

24. Line 287: "Considering the height of the Planetary Boundary Layer (PBL), we use the ERA5 wind at 500 m above the ground (Figure 3c)" I don't understand what you mean by that. The PBL height changes during the day and year. How is 500 m related to the PBL height? In your abstract you give the emission rates for a height of 330 m.
The enhancement in $XCH_4$ that is being used to estimate the emission rates is composed of $CH_4$ molecules that have been advected in different heights. To me it is unclear why a certain height should be more representative than the other (at least in the PBL). Shouldn't an average wind speed over the entire vertical spread of the PBL be used. This would of course massively increase the uncertainty of your estimation. Please comment.

25. Figure 9: The grids that are shown are a superposition of the days on which, in the daily average, the wind blew in the respective narrow-regimes, right? If so, this should be explained once more in text in section 3.3.2.
I don't understand where all the missing data points originate from. If for example the cone-plumes are never advected into the narrow wind-regime at NW-SE, then, for the respective grid, the calculation is 0-0=0, isn't it? If so, you can of course filter these grids to make a distinction to cases where $XCH_{4NW}$ and $XCH_{4SE}$ are equal but not zero. If this is being done please explain it somewhere in the text.

26. Line295 ff: „The final estimated emission strength is weighted by the number of the valid binning data in the plume maps under different wind regimes (i.e. 171 for narrow NE-SW and 26 for narrow NW-SE, respectively)." I do not understand the weighting. Are there 171+26 days in total? The emission rate of 9.8E26 molec./s from NE-SW regime is being weighted with 171 days and the 14.0E26 molec/s with 26 days. Result is then 10E26 molec/s which is given in line 303? If I understand correctly please insert the information that by "number of the valid binning data" you mean "number of days on which, on average, the wind blew in the respective wind-regime."

27. Line 352 ff: "However, their speeds decrease by 19% at 10 m and increase by 32% at 500 m, which results in higher emission rates by -23% and 13 %, respectively." How can that be? Wind-speed is linear in the calculation of ε, isn't it? Accordingly, the emission rates should also be -19% & +32%. Please comment.
Furthermore, "higher emission rates" is not correct for describing a decrease and an increase. Please rephrase.

**Technical comments**

28. Please consider perceptually uniform sequential colormaps, especially for figures 2, 4 and 6. Diverging colormaps are helpful in displaying differences, which in your case would only make them suitable for plotting wind-assigned anomalies and XCH4 enhancements. If you stick to the red-blue diverging colormap for the anomalies consider hatching the grids with missing data. At the moment they are easily mistaken for value 0.
    If you are using python to generate plots you might have a look here:
    https://matplotlib.org/2.0.2/examples/color/colormaps_reference.html

29. For many citations there is a dot missing after "et al".

30. For almost all figures the labeling is way too small. Please increase the font size corresponding to the text.

31. Line 12: „Intensive coal mining activities are in the Upper Silesian Coal Basin (USCB) in southern Poland, resulting in large amounts of methane (CH4) emissions." Maybe shift the "are" in front of "resulting".

32. Line 13: "Annual CH4 emission reached to 448 kt according to the European Pollutant Release and Transfer Register (E-PRTR, 2017)." Please remove the "to" or change to "… reached up to 448 kt …"

33. Line 14-15: "As a $CH_4$ emission hot spot in Europe, it is of importance to investigate its emission sources and accurate emission estimates". Maybe insert "make" in front of "accurate emission estimates"

34. Line 16: "column-averaged dry-air molar fraction observations of CH4". Please change to "mole fraction observations".

35. Line 16-20: It is a rather long sentence. Maybe split up.

36. Line 27 ff: "… with using the Carbon dioxide and Methane (CoMet) inventory …" What information is actually used from the CoMet inventory? As you report your emission estimates in the next sentence I assume that, here, you just take the locations of the shafts. Please be more specific, as the CoMet inventory also reports emission rates of individual shafts.

37. Line 28: Not sure what is meant by "performed"? An inventory is not performed. How about "… from 2018", "… covering the year 2018", "issuing the year 2018" or something equivalent.

38. Line 34-35: "When using different wind coverage and different wind segmentation, an uncertainty of 4.2% and -2.1% is obtained, respectively". How is an uncertainty negative? Maybe uncertainty is not the adequate word.

39. Line 40-42: This sentence is hard to read. In my opinion the word "and" is used to often. I think in "… and waste disposal …" you can remove it.

40. Line 43: „… to the atmosphere CH4 level are still …". This seems off. Maybe change to "atmospheric" or "atmosphere's"

41. Line 75: "… data sets provide a large coverage and long-term XCH4/TXCH4 observations, which helps to better estimate CH4 emission …" I guess it should be "help", not "helps".

42. Eq. 1: The square root should also include the numerator. The calculation of the standard deviation is trivial. If you want you can remove the equation.

43. Line 111: "… emissions from ships with a magnitude of 19 are much lower …". What do you mean by "magnitude"? Do you mean the "count" of ships?

44. Line 113: "Compared to its high amount, the seasonal variations of the fugitives sector can be ignored." Sounds off to me. Maybe avoid "high amount" when referring to emission rates.

I suggest "The seasonal emission variations of the fugitive sector are minor and can be ignored" or similar.

45. Line 113 – 119: From here on, the paragraph is no longer stringent to me. What the authors are basically saying is that the fugitive sector is dominant in the USCB. As the fugitive sector has minor seasonal variations they do not consider them. I suggest the following restructuring from line 111 onwards: "Thus, these three sectors are not shown here. The sources from agriculture livestock (1.7E25 ± 4.0E25 molec./s) amount only 4% of the total emissions in this region. The dominant CH4 sources in this region are fugitive sources from from energy production and distribution (e.g. fuel use). With a mean value of 7.9E26 molec./s and a standard deviation of 2.2E25 molec./s they account for 82% of the anthropogenic CH4 emissions in the CAMS-GLOB-ANT inventory (9.7E26 molec./s in total). This becomes particular visible in the spatially overlapping distribution within the USCB (see Figure 2). The seasonal emission variations of the fugitive sector are minor and can be ignored. Therefore, we apply the three-year mean of total emissions at grids with significant emissions without considering seasonal variations in the simple plume model (see Sect. 2.3)"

46. Figure 1: The coloring is highly unfortunate. In the legend the "Fugitives" is listed last and easily mistaken for "Off road transportation". Please list "Fugitives" first and change the color for "Off road transportation".

47. Figure 2: I assume that the barely visible gray lines are the borders to the Czech Republic and Slovakia. Please increase the resolution of the basemap so that the borders can be recognized as such. For a better orientation you might consider inserting country abbreviations.
At first glance, the two heatmaps look identical, which is of course the point being made here. However, I'm a bit unsure about the gain of information when two nearly identical images are shown side by side. Perhaps a heatmap of the percentage shares of fugitive emissions compared to overall anthr. emissions would be better. Please comment.

48. Line 146: Comma before "which"

49. Line 147: "… it is able …" What does "it" refer to? I guess it refers to the combined product, which you introduce as such only in the following sentence. I suggest "… we are able to …".

50. Line 154: You reference the figure 2 from a previous publication. In my opinion it would be beneficial to actually show the figure again.

51. Line 159 and Eq. 2: The indices i of $(x_i, y_i)$ should be subscripted.

52. Figure 3: The individual plots in Figure 3 will separately be referred to with a, b and c (e.g. Line 288). Please add a numbering to the plot or change reference in the text to left, middle and right.

53. Figure 4: Please increase the size of the squares for the CAMS-GLOB-ANT sources. The color of the sources is so difficult to distinguish. An increase in the size of triangles for the CoMet sources would also be beneficial, although this might be more difficult as the triangles overlap. If possible, please improve the visibility. If not, you might consider providing a zoom to the shafts in a separate subsection, which was suggested by Referee#1.

54. Figure 4: In the caption it says "during November 2017-December 2020". Does this mean the displayed XCH4/TXCH4 data are an average of this period? If so, please indicate this in the caption. Otherwise, please specify the displayed day.

55. Figure 4: I assume that the white grids are missing data. Please indicate this in the caption. Moreover, the color choice is unfortunate, as it is missing values are difficult to distinguish

from the mid-range values in the colorbar. Please see my earlier comment regarding colormaps.

56. Figure 4: For a better comparison please consider using an identical colormap for a) and b). The TROPOMI & IASI data product has of course higher values. If the colorbar consists of the same colors, please indicate the shift in values in the caption.

57. Figure 5 & 7: If a diverging colormap is being used, please center the colorbar to the value 0. Please use the same colormap for all four plots.

58. Figure 5 & 7: Please avoid the term "anomalies" if you are not referring to wind-assigned anomalies. Rather use "enhancement" as suggested in an earlier comment

59. Figure 5 & 7: Please do not repeat the identical title for multiple plots in the figure. I Suggest to name the lines on the left with [XCH4, TXCH4].T. Name the columns with [CAMS, modelled (cone-plumes + ERA5), correlation plot]. Instead of "modelled (cone-plumes + ERA5)" you could of course choose a term of your choice. Something like "wind-assigned anomalies (SW-NE)" or similar would be fine too.

60. Figure 5 & 7: The colorbar-label for the left plots (a & d) and the middle plots (b & e) are currently the same. The left plots are displaying XCH4 enhancements (i.e. XCH4 – background), the middle plots are displaying wind-assigned anomalies. Please correct the colorbar labels.

61. Figure 5 & 7 caption and title of middle plots: "… the wind-assigned anomalies (NE-SW) …" Shouldn't it be "SW-NE"? Otherwise the positive values should be in the NE.

62. Figure 5 & 7 correlation plots (c & f): Please remove the title. The information is already given in the axis' labels. Also, as mentioned before, the use of ΔXCH4 is not consistent.

63. Line 216: "9.1E24 ± 1.2E24 molec./s" I guess there is a typo in the exponent.

64. Line 227: "Figure 6 illustrates the enhance XCH4 (raw XCH4-background in the upwind) distribution …" Please correct "enhance" to either "enhanced XCH4" or "XCH4 enhancement". Why is "in the upwind" specified? From the explanation in the appendix of your earlier publication the background determination is not limited to the upwind.

65. Line 244: "… anomalies show high amounts around the areas …" To me "amounts" sounds off. Please consider something like "high concentrations", "high methane content" or something similar.

66. Figure 8: Please be precise in the labeling of the horizontal lines, i.e. "Total Emission (CAMS-GLOB-ANT", "Total Emission (CoMet inventory)".

67. Figure 8: Please remove the shaded background and instead add a legend: "a priori: squares CAMS-GLOB-ANT, triangles CoMet inventory", or something similar.
If plotted among each other triangles and squares are easier to compare.

68. Figure 8: The error bars are very small, as you mention in the caption. Nevertheless, please use either a uniform color, such as black or gray, or simply the color of the respective marker. At the moment it seems like they change colors randomly.

69. Line 280: „Here we investigate the wind uncertainties …". Please insert a comma after "here"

70. Line 284: "Compared to the wind at 330 m, the wind distributions are similar …" Please specify, in the whole text, that you are referring to the distribution of wind directions.

71. Section 3.3.2: Since the designation SW and NE were used previously and now SW and NE are still used for narrow, the text is a bit confusing. Either _narrow is always subscripted consequently, as is being done in the caption of Fig. 9, or, alternatively, the subscripts $SW_{1/2}$ or $SW_{1/4}$ could be used to specify whether the wind field is divided into halves or quarters.

72. Figure 9 and text in section 3.3.2: Isn't it "SW-NE" instead of "NE-SW"?

73. Line 310: "The wind category here is based on its predominant wind fields over the USCB region …". Please change "its" to "the" or rephrase.

74. Line 311: "To investigate its uncertainty, we apply another kind of segmentation:" What does the "its" refer to? Please change to "To investigate the effect of the segmentation on the uncertainty in the emission rate estimation, we additionally apply another kind of segmentation" or similar.

75. Line 335: "To investigate the CH4 emissions from this hot spot, the CoMet campaign was performed in 2018. Locations and emission rates of the ventilation shafts of the coal mine used in this study are based on this inventory". "This" probably refers to the CoMet campaign. A campaign is not an inventory. Please rephrase.

76. Line 340: "… and reasonablely compared to the CoMet inventory (6.6E26 molec./s)" Please change "reasonablely" to "reasonable"

77. Line 343: "… up to 5.68E26 molec./s derived from one flight (Kostinek et al.(2021)). Similar 2D anomalies and plumes are also observed …" Similar to Kostinek et al.? Otherwise, please separate into two paragraphs to make it clear that you are now writing about plumes/anomalies and no longer about total emission estimates.

78. Table A-2: Instead of "CAMS emission" & "shafts emission" I think it would be better to use " CAMS-GLOB-ANT" and "CoMet inventory" according to the caption. In the left column you could label the line as "prior emission sources" or similar.

79. WMO Reference from Line 40 is missing.

---

## Author Comment (AC1)

**Response to Referee #1**

We would like to thank reviewer #1 for taking the time to review this manuscript and for providing valuable, constructive feedback and corresponding suggestions that helped us to further improve the manuscript.

In this author's comment, all the points raised by the reviewer are copied here one by one and shown in blue color, along with the corresponding reply from the authors in black.

This paper estimates the methane ($CH_4$) emissions from one of the most outstanding $CH_4$ sources in Europe using a multi-platform of reference data sets (space-based observations, atmospheric simulations, emission inventory) and a novel, robust, simple approach. The paper provides new and interesting findings, and is written and structured well; therefore, I suggest it to be suitable for publication in ACP after specific and technical comments (listed below) are addressed.

**1. Specific comments:**

1.1 Title: The title suggests that the $CH_4$ emission quantification is jointly done using TROPOMI, IASI, and CAMS products. However, the CAMS data was mainly used as a validation tool of the wind-assigned anomaly method. Section 3.2 indeed shows and discusses briefly an example day using the CAMS and space-based observations, and Figure 8 summarizes the $CH_4$ emission rates using all different data sets, but this figure is not discussed in the text. I would recommend to change it to "Quantifying hard coal mines $CH_4$ emissions from TROPOMI and IASI observations, high-resolution CAMS forecast data and the wind-assigned anomaly method".

(1) We would like to thank the referee for pointing this out. Yes, the CAMS data were used to evaluate our method and helped to choose the most suitable wind information. However, the CAMS data (forecast and inventory) are not used in estimating the emission strengths from the TROPOMI and TROPOMI+IASI products. The high-resolution CAMS forecast data are considered as supporting information and thus, we would like to keep "using" in the title.

(2) We have added some discussion related to Figure 8 (see 3.4 below).

**2. Section Data sets and method:**

2.1. A subsection describing the USCB region would help the reader, for example, including the orography, the predominant wind regimes, etc. In addition, given that the COMet inventory is used in this paper, I would also recommend including a subsection providing some details about it.

Thanks for this comment. We have added related information to the introduction (in the 2nd and 3rd paragraph, respectively) as the referee suggested:

"The USCB is in the Silesian Upland, which is a plateau between 200 and 300 m above sea level with a predominant south-west wind. The USCB within Poland covers an area of over 5800 km$^2$, and to its south is the Tatra Mountain ridge with elevations larger than 2000 m a.s.l."

"A variety of state-of-art instruments, including in situ and remote sensing instruments on the ground and aboard five research aircraft, were deployed in order to provide independent observations of GHG emissions on local to regional scale and provide data for satellite validation."

**2.2 Line 100: Include some reference and explanation about the expected uncertainties of the CAMS-GLOB-ANT inventories.**

The reference has been added.

The CAMS inventories (anthropogenic and natural emissions) do not provide estimates of the uncertainties and a potential work on the uncertainty estimates might be available in the future (we acknowledge Dr. Claire Granier from Laboratoire d'Aerologie, Toulouse, France for providing this information).

**2.3 Line 135-136: Include information about the TROPOMI overpass (time, frequency,...) similar to IASI.**

The information about the TROPOMI overpass has been added:

"The instrument crosses the equator at about 13:30 local solar time at each orbit with a repeat cycle of 17 days. It observes a full swath (2600 km) per second with an orbit duration of 100 min."

**2.4 Line 138: Include information about the number of quality-filtered TROPOMI dataset (and also for the combined TROPOMI+IASI product in the next paragraph). Is the space-based data set robust enough for CH$_4$ emission estimates?**

(1) The number of data points in the quality-filter TROPOMI dataset is about 16,000 over three years. About 12000 data points are collected from the TROPOMI+IASI product. We have added this information to the text.

(2) TROPOMI XCH$_4$ data have been characterized by high spatial- and temporal-resolution with being in good agreement with TCCON ($-3.4 \pm 5.6$ ppb) and GOSAT ($-10.3 \pm 16.8$ ppb) (Lorente et al., 2021). TROPOMI XCH$_4$ has been used in different studies to detect and estimate the CH$_4$ emissions from different sources, e.g., from coal mining (Varon et al., 2020), and from the oil and gas sector (Pandey et al., 2019; De Gouw et al., 2020). Moreover, in a previous study (Tu et al., 2022) the emission strength derived from TROPOMI was compared to one-day observations of ground-based FTIR instruments and both have the same order of magnitudes. Our result derived from TROPOMI products in this study is close to the CoMet inventory and results of other studies by using different methods. Thus, the space-based data set is robust enough for CH$_4$ emission estimates.

**2.5 Line 144: Include reference for the improvement of IASI on the NWP systems.**

The references below have been added to the text:

Collard, A. D.: Selection of IASI Channels for Use in Numerical Weather Prediction, ECMWF, https://www.ecmwf.int/node/8760, 2007.

Coopmann, O., Guidard, V., Fourrié, N., Josse, B., and Marécal, V.: Update of Infrared Atmospheric Sounding Interferometer (IASI) channel selection with correlated observation errors for numerical weather prediction (NWP), Atmos. Meas. Tech., 13, 2659–2680, https://doi.org/10.5194/amt-13-2659-2020, 2020.

2.6 Line 145: Is the statement about "different atmospheric trace gas profiles" referring to only $CH_4$ or to all the MUSICA products? If the latter, please consider including other references for completeness such as Schneider et al. (2022), Dieckmann et al. (2021) or García et al., (2018).

Thanks. The statement is a general introduction about IASI and adding other references as recommended by the referee is better.

Diekmann, C. J., Schneider, M., Ertl, B., Hase, F., García, O., Khosrawi, F., Sepúlveda, E., Knippertz, P., and Braesicke, P.: The global and multi-annual MUSICA IASI {H2O, δD} pair dataset, Earth Syst. Sci. Data, 13, 5273–5292, https://doi.org/10.5194/essd-13-5273-2021, 2021.

García, O. E., Schneider, M., Ertl, B., Sepúlveda, E., Borger, C., Diekmann, C., Wiegele, A., Hase, F., Barthlott, S., Blumenstock, T., Raffalski, U., Gómez-Peláez, A., Steinbacher, M., Ries, L., and de Frutos, A. M.: The MUSICA IASI $CH_4$ and $N_2O$ products and their comparison to HIPPO, GAW and NDACC FTIR references, Atmos. Meas. Tech., 11, 4171–4215, https://doi.org/10.5194/amt-11-4171-2018, 2018.

Schneider, M., Ertl, B., Diekmann, C. J., Khosrawi, F., Weber, A., Hase, F., Höpfner, M., García, O. E., Sepúlveda, E., and Kinnison, D.: Design and description of the MUSICA IASI full retrieval product, Earth Syst. Sci. Data, 14, 709–742, https://doi.org/10.5194/essd-14-709-2022, 2022.

2.7 Line 151: Some information about the improvements/differences of the wind-assigned anomaly method with respect to other top-down approaches would help the reader to have a better idea of novelty and benefit of this method.

There are generally two kinds of methods to estimate the $CH_4$ emission strengths. The first method is based on the atmospheric transport model (e.g., GEOS-Chem), which is considered as a forward model to create the relationship between $CH_4$ and surface emissions (Zhang et al., 2020). The optimization is the inversion step to obtain the best fit between the observations and the model. This method is mostly used on regional to large scales. Another method is based on the conservation of mass (e.g., divergence), i.e., the sum of the emission and background equal to the observations. This divergence method was first used to estimate $NO_2$ emissions (Beirle et al., 2019) and later extended to estimate $CH_4$ emissions (Liu et al., 2021).

Our wind-assigned method is based on the theory of conservation of mass and uses a simple cone plume model, which is easy to apply than the other methods and the estimated emission strengths are reasonable compared with the ones from other studies. This information has been added to the text.

Beirle, S., Borger, C., Dörner, S., Li, A., Hu, Z., Liu, F., Wang, Y., & Wagner, T. (2019). Pinpointing nitrogen oxide emissions from space. Science Advances, 5(11). https://doi.org/10.1126/sciadv.aax9800.

Liu, M., van der A, R., van Weele, M., Eskes, H., Lu, X., Veefkind, P., et al. (2021). A new divergence method to quantify methane emissions using observations of Sentinel-5P TROPOMI. Geophysical Research Letters, 48, e2021GL094151. https://doi. org/10.1029/2021GL094151.

Zhang, Y., Gautam, R., Pandey, S., Omara, M., Maasakkers, J. D., Sadavarte, P., Lyon, D., Nesser, H., Sulprizio, M., P., Varon, D., Zhang, R., Houweling, S., Zavala-Araiza, D., Alvarez, R. A., Lorente, A., Hamburg, S. P., Aben, I., Jacob, D.: Quantifying methane emissions from the largest oil-producing basin in the United States from space. Science Advances, 6(17), eaaz5120. https://doi.org/10.1126/sciadv.aaz5120, 2020.

2.8 Line 176: Describe slightly the results obtained (first validation of the wind-anomaly method) in Madrid experiment (Tu et al., 2021) to highlight the robustness and reliability of the method.

Thank you for this important point. We have added more information in section 2.3:

"This method was firstly used to estimate the $CH_4$ emission from landfills in Madrid, Spain based on nearly three-year space-borne $XCH_4$ data, and different opening angles were investigated to obtain an empirical value (60º) (Tu et al., 2022). The $CH_4$ emission strengths derived from satellite products have the same orders of magnitude as the ones from single-day observations by ground-based instruments, showing that this method works properly."

**3. Results and Discussion:**

3.1 Line 205: During the COMet campaign, high-resolved aircraft profiles were performed allowing $CH_4$ emission rates to be estimated (e.g. Fiehn et al., (2020), Kostinek et al. (2021)). Have the authors analyzed the aircraft dataset to corroborate that the wind fields at 300 m are the optimal option? As discussed in the "Uncertainty analysis", the vertical wind shear is the most critical factor to estimate the $CH_4$ emission rates.

We did not analyze the short-term aircraft dataset in this study. Our method is based on a long-term dataset, i.e., the CAMS $XCH_4$ and wind-assigned method to find out that the estimated emission strength fits best with the CAMS-GLOB-ANT inventory by using wind information at 330 m. There might have high biases for only using a short-term period of data. Moreover, the wind speed at 330 m is more or less an average of the ones at 10 m and 500 m.

3.2 Line 206-209: Please provide more details about this statement (ie, the small changes in wind could not be properly captured by ERA wind fields). What would the net effect of ruling out these pixels be in the total estimations?

We consider the wind changes (speed and direction) over daytime and these effects are averaged based on the time scale and super-positioned for all emission sources. The enhanced column is proportional to distance, and it is set to zero only when the distance is zero, i.e., the points locate exactly in the emission sources' places. The distance-related filter ("the points whose distances to the nearest dominant sources are less than 10 km") is not applied in calculating the enhanced columns and in estimating the emission strengths. The previous correlation plots in the manuscript were distance-related filtered, which might mislead the readers. This sentence is removed, and correlation plots have been updated.

3.3 Figure 5: Why is there more scatter in the positive anomalies?

These positive anomalies represent the values in the SW area (i.e., the downwind region of the NE wind), where more emission sources are located than in the NE area. The enhanced $CH_4$ columns (Eq.

2) are proportional to the distance, and thus, the positions that are near the emission sources can be easily affected by the sources. Although we removed the points whose distances to the nearest dominant sources are less than 10 km in the previous correlation plots, the points might be affected by other sources, which probably results in more scatters in the positive anomalies. The figure below shows that most scatters are related to the points that are near the emission sources.

[Figure]

3.4 Line 224: As mentioned before, section 3.2 shows and discusses briefly an example day using the CAMS and space-based observations, but the emission rates using the whole data set is not included and discussed. If I understand well, the analysis was done because Figure 8 summarizes the $CH_4$ emission rates using all different data sets for the discussion of effect of wind at different levels, but this figure is not discussed in the text (neither in section 3.2 and section 3.3). Including this information in the text would help to compare the results with COMet inventories (discarding the influence of space-based observations uncertainties).

Thank you for this important comment. The following paragraph has been added to the paper:

"Figure 8 summarizes the estimated emission strengths derived from different products based on different a priori knowledge of inventories and wind information at different altitudes (for specific values see Table A- 1). Different a priori inventories result in 16%-32% changes in strength at different altitudes, which is generally smaller than the 47% difference in the total amount of inventories (9.7E26 for CAMS-GLOB-ANT and 6.6E26 molec./s for CoMet inventory). This is probably due to the different locations of sources and different proportions of each emission source in the total strengths in the two inventories. When using the CAMS-GLOB-ANT inventory, $CH_4$ emission rates derived from CAMS $XCH_4$ and $TXCH_4$ are ~37% and ~56% higher than those derived from TROPOMI $XCH_4$ and IASI+TROPOMI $TXCH_4$, respectively. This difference is mainly due to the difference between the CAMS forecast and satellite products. The strength increases with respect to the increasing wind speed at higher altitude. Whereas the increment is not always proportional to the wind speed, i.e., less increase in the strength with respect to the wind speed at higher altitude (see Sect. 3.3.1)."

3.5 Line 247: There is a significant change of slope for the combined $TXCH_4$ product (Figure 7 f). Do the authors have some explanation for this?

The different slope for modeling emission strength derived from $TXCH_4$ products is mainly due to the difference between $XCH_4$ and $TXCH_4$. $XCH_4$ is the ratio of the total column of $CH_4$ and the total

column of dry air, whereas the TXCH₄ is the ratio of the total column of CH₄ in the troposphere and the column of the tropospheric dry air. Mixing ratios of CH₄ decrease in the stratosphere, resulting in higher absolute values of TXCH₄ than XCH₄, with a slope of 1.07 (see figure below). The modeled ΔXCH₄ and ΔTXCH₄ in Figure 7(c) and (f) are the same product and thus, a lower slope is expected in fitting the TROPOMI+IASI ΔTXCH₄ to the model ones. The ratio (1.07) of TXCH₄ and XCH₄ is close to the ratio (1.05/0.89=1.18) of the slopes in Figure 7 (c) and (f), which further supports the explanation above.

[Figure]

**4. Technical comment:**

4.1 Line 19 and line 76: Include the period covered by this study in the abstract and introduction.

Corrected, thanks.

4.2 Line 70: Include acronym for tropospheric XCH₄ (TXCH₄).

Thanks, this has been added.

4.3 Line 89: Consider plural for "aerosol".

corrected, thanks.

4.4 Figure 1: The colours used for "Off Road transportation" and "Fugitives" are quite similar and make it hard to distinguish them only by looking at the plot. The final full stop is missing.

The figure has been updated.

[Figure]

Figure 1: Stacked area plot for different sectors of the monthly averaged CAMS global anthropogenic emissions (>1E20 molec./s) in the USCB region for 2017-2020 (https://permalink.aeris-data.fr/CAMS-GLOB-ANT, last access: 22 December 2021. Granier et al., 2019).

**4.5 Line 154: Please consider moving the description of the ERA wind model to line 164.**

This has been done as the referee recommended.

**4.6 Line 166: 08:00 UTC or 09:00 UTC as in the CAMS products description. Why do not use the CAMS products starting at 08:00 UTC?**

The daily wind-assigned plume from each emission source is averaged over daytime (8:00 UTC – 18:00 UTC), i.e., we considered the wind changes over the day. The different single-source-resolved plumes from all emission sources are super-positioned to a total daily plume. We then fit the different daily plumes to the CAMS $XCH_4$. Because the daytime average emissions are calculated, we then use the daily averaged CAMS $XCH_4$ as well. However, the CAMS $XCH_4$ has a temporal resolution of 3h, starting from 00:00 UTC. Therefore, the CAMS $XCH_4$ at 9:00, 12:00, 15:00, and 18:00 UTC are used to calculate the daily average, and their standard deviations are considered as uncertainties.

**4.7 Line 180: Correct "500 m" and "three-year average" in the figure caption.**

Thanks, corrected.

**4.8 Figure 3: Correct "TROPOMI" in the figure caption.**

The typo in Figure 4 has been changed accordingly.

**4.9 Figure 5: Include the meaning of the error bars in the figure caption (is the STD given by Eq 1?).**

The information has been added.

**4.10 Figure 6: To be consistent with the other figures, please consider modifying this figure accordingly (coloured bars, labels (a, b, c), "modelled" in the title of third subplot, definition of first subplot,...)**

Thanks, the figures have been modified.

[Figure]

Figure 6: ΔXCH$_4$ together with the ERA5 wind at 12:00 UTC from (a): TROPOMI observations at 11:34 UTC, (b): CAMS forecast at 12:00 UTC, and (c): from the simple plume model (averaged over the daytime) based on the CAMS-GLOB-ANT inventory over the USCB region on an example day (6 June 2018). The "bg" in the title of (a) and (b) represents the average background, derived from the mean XCH$_4$ in the upwind region (50.3º-50.8º N, 19.5º -20.0º E).

**4.11 Figure 7: Correct "TXCH$_4$" in subplot (e).**

Thanks, corrected.

**4.12 Figure 8: Correct "wind" in the x-label for 300 m. Correct "300 m, 500 m".**

Changed accordingly.

**4.13 Line 284: Correct Figure A-1 to plain text.**

Changed accordingly.

---

## Author Comment (AC2)

**Response to Referee #2**

We would like to thank the reviewer #2 for taking the time to review this manuscript and for providing valuable, constructive feedback and corresponding suggestions that helped us to further improve the manuscript.

In this author's comment, all the points raised by the reviewer are copied here one by one and shown in blue color, along with the corresponding reply from the authors in black.

**General comments:**

The manuscript „Quantifying hard coal mines $CH_4$ emissions from TROPOMI and IASI observations using high-resolution CAMS forecast data and the wind-assigned anomaly method" by Qiansi Tu et al., reports on a top-down approach to estimate methane emissions on the region scale. In their work the authors focus on the $CH_4$ emissions from hard coal mines in the Upper Silesian Coal Basin. Their emission estimation is based on applying a simple cone-plume-model and fitting the associated wind-assigned anomalies to enhancements in the $XCH_4$ data retrieved by satellite observations form TROPOMI and IASI over a period of three years. Simple, straight forward to apply approaches, as presented by the authors, to estimate $CH_4$ emissions on a local scale are highly relevant, especially in the light of recent COP 26 and the Global Methane Pledge that emerged from it.

The manuscript is well structured, but poorly written. The high amount of technical errors suggests that the authors have made an insufficient effort in proofreading, before submitting their manuscript to the journal. Nonetheless, I recommend the study as suitable for publication in ACP after the major revision has been addressed.

Thanks for pointing this out. We will do our best again to improve the language.

**Specific comments:**

1. molec/s is a rather small-scaled unit for an observation-period of three years over a 100x100km region. Maybe kt/year is more suitable. Would also get rid of hard to read exponential nomenclature. Furthermore, in the introduction your use of units switches from kt/yr to $TgCH_4yr-1$.

   The units of molec./s and $TgCH_4yr-1$ are changed to kt/year as the referee recommended. The emission rate used in the cone-plume model has a unit of molec./s as typically used for remote sensing application for corresponding with column amounts (molec./area unit), so we would like to keep this unit in the text as well.

2. Sometimes it is not clear whether the CAMS GHG dataset or the CAMS emission inventory is being referred to (e.g. Lines 188 and 206; title of Fig. 5; caption Fig. 9; ...). This can be particular challenging, when the authors' emission estimates retrieved from the CAMS GHG dataset are compared to the CAMS emission inventory. I recommend using "CAMS-GLOB-ANT" throughout the text whenever referring to the emission inventory.

Thanks, we have changed "the CAMS emission inventory" to "CAMS-GLOB-ANT" to make it clearer.

3. Line 21: "wind directions" Throughout the text, the division into wind regimes is designated differently. I think the designations wind regimes/segments/divisions are suitable, wind sectors/sections are not. Please adjust the text accordingly.

Thanks, the "wind sectors/sections" are changed to "wind regimes".

4. Line 154: "This model is referred to as simple plume model". A lot of plume models can be described as "simple". I suggest referring to it as "cone plume model". This would have the advantage that the designation is self-descriptive. If the authors want to stick with the term "simple plume model" for consistency with the earlier publication, that's fine with me.

The "cone plume model" is appropriate to represent the characteristic of our method. Changed accordingly.

5. Line 162: To assume constant emission rates for three years is rather bold. According to the E-PERTR variations of a few percent are to be expected. Since the uncertainties in your estimate of emission rates are rather small, much of this might be due to interannual fluctuations. Please reconsider this statement and consider estimating emission rates for individual years.

The yearly emission strengths from CAMS-GLOB-ANT over the study area are 802.0 kt/year, 803.6 kt/year, and 807.4 kt/year in 2018, 2019, and 2020, respectively. These changes are 0.2% from 2018 to 2019, and 0.5% from 2019 to 2020, which is low and can be neglected.

The CAMS data are collocated to the TROPOMI+IASI data (see 11[th] comment), so the yearly amount of data is less and thus, poorer correlations of wind-assigned anomalies derived from the CAMS and the cone plume model are found:

[Figure]

The corresponding estimated emission rates are $1597 \pm 92$ kt/year, $689 \pm 118$ kt/year, $512 \pm 12$ kt/year (i.e., $1.9E27 \pm 1.1E26$, $8.2E26 \pm 1.4E26$, and $6.1E26 \pm 1.4E25$ molec./s) in 2018, 2019 and 2020, respectively. Higher uncertainties are found for the years 2018 and 2019. Therefore, using data for longer time periods results in better correlations and lower uncertainties.

6. In Eq. 2 you introduced $\Delta XCH_4$ as the enhanced $CH_4$ column. Later, especially in figures, you use $\Delta XCH_4$ for the wind-assigned anomalies. Please introduce a distinctive notation for the wind-assigned anomalies to avoid confusion with the enhanced $CH_4$. See also the next comment for that.

We would like to thank the referee for pointing this out. The $\Delta XCH_4$ in this study is only used to represent the enhanced $XCH_4$. The $\Delta XCH_4$ used for the wind-assigned anomalies has been changed accordingly.

7. Line 163: Where does the empirical value of 60° come from? From your earlier publication I know that it comes from TROPOMI $NO_2$ measurements, but this should be explained and cited here.

The sentence has been changed as the referee recommended:

"$\alpha$ is the angle of the emission cone and has an empirical value of 60°, which has been derived from TROPOMI $NO_2$ measurements (Tu et al., 2022)"

8. (1) Line 169-170: „For each wind sector, an averaged plume is computed and the difference of the two plumes are therefore the wind-assigned anomalies". Above it is said that daily averaged plumes are calculated. Here, "averaged plume" refers to a plume averaged over all daily-plumes, which propagate NE/SW on daily-average. That means at that step you have two plumes for the entire three years. At each pixel your wind-assigned anomaly is calculated by $\overline{XCH_4}_{SW}(i,j) - \overline{XCH_4}_{NE}(i,j)$. Without checking your previous publication, I couldn't understand this. A few more equations would be helpful to explain your approach. Something like: $\overline{XCH_4}_{SW/NE}(i,j) = \frac{1}{N_d}\sum_d^{N_d} XCH_{4,d}(i,j)$ with $N_d$ = number of days and i,j SW/NE and wind-assigned anomaly = $\overline{XCH_4}_{SW}(i,j) - \overline{XCH_4}_{NE}(i,j)$.

(2) Please consider showing table A-1 here instead of the appendix.

(1) We have added the requested explanation in section 2.3 of the paper:

The daily plume from each point source (location at (i,j)) is averaged over the daytime (8:00 UTC - 18:00 UTC):

$$\overline{XCH_4}_{(i,j)} = \frac{1}{11}\sum_{t=1}^{11} XCH_{4(i,j),t} \qquad \text{Eq. 3}$$

these daily plumes are super-positioned over all point sources to obtain a daily plume ($\overline{XCH_4}_{daily}$):

$$\overline{XCH_4}_{daily} = \sum_{s=1}^{N_s} \overline{XCH_4}_{(i,j),s} \qquad \text{Eq. 4}$$

where $N_s$ represents the number of the sources.

The wind distributions at different height levels (10 m, ~330 m, ~500 m) over the USCB region are presented in Figure 3. The wind speed increases with increasing altitude (see Table 1). The ERA5 wind is divided into two opposite wind regimes based on directions (e.g., 135°-315° for SW and the rest for NE). For each wind regime, an averaged plume is computed:

$$\overline{XCH_4}_{SW/NE} = \frac{1}{N_d}\sum_{d=1}^{N_d} \overline{XCH_4}_{daily,d}$$

Eq. 5

where $N_d$ is the number of the days with SW wind or NE wind.

The difference between the two plumes is therefore the wind-assigned anomalies:

$$wind-assigned\ anomalies = \overline{XCH_4}_{NE} - \overline{XCH_4}_{SW}$$

Eq. 6

(2) The Table A-1 has been moved to section 2.3 as the referee recommended.

9. Line 175: Background removal is critical for correctly estimating the emission rate. From your earlier publication I know that the uncertainty in the background subtraction is included in the uncertainty of your enhanced $XCH_4$ values. Please add a short statement about uncertainty in background subtraction in chapter 3.3. Moreover, I would be interested in how the background differs between CAMS and TROPOMI data.

A short statement about the uncertainty in background subtraction has been added:

"$CH_4$ signal is weak compared to the background concentration which shows an increasing trend with obvious seasonality and strong day-to-day signals. It is necessary to remove the background signals before estimating the emission strengths. However, the imperfect elimination of the background introduces uncertainties, which can be determined by considering the deficits of the background model and the noise in the background (Tu et al., 2022). In this study, the uncertainties of the estimated strengths include the background uncertainties."

The correlations of $XCH_4$, background, and the enhancement (raw $XCH_4$ - background) between CAMS and TROPOMI are shown below:

[Figure]

10. Line 195: "The $XCH_4$ anomalies (raw-background) and the wind-assigned anomalies are presented in Figure 5a and b, respectively". Please change to $XCH_4$ enhancement, to avoid risk of confusion to the wind-assigned anomalies. This would be more consistent as the term "enhanced" has already been used for background-free $XCH_4$ in the context of Eq.2.

Thanks. These have been changed.

11. Line 196-197: I don't understand what you mean by: „Note, that the CAMS XCH$_4$ is coincided with TROPOMI XCH$_4$ for better comparison". Were the CAMS data interpolated to the TROPOMI grid and accordingly the filtered TROPOMI grids are also missing for CAMS? But why are different grids missing for all plots displaying wind-assigned anomalies?

Thanks for pointing this out. There are nearly 400,000 data points for the CAMS XCH$_4$ during the study period over the study area due to its high spatial resolution (0.1° × 0.1°). It is hard to make the data processing program run for this high amount of data. Apart from that, the CAMS XCH$_4$ is used as an evaluation of our method in this study and to find the best ERA wind which is used for TROPOMI and TROPOMI+IASI data later. Thus, for each TROPOMI measurement, the nearest CAMS XCH$_4$ is selected as the co-located data, i.e., both data sets have the same amount of data (16,553). However, the TROPOMI data is further collocated with the IASI data (TXCH$_4$), which results in a smaller data set (12,354). The different amounts of data lead to different grids missing.

To make the data sets to be consistent, we co-locate the CAMS to the TROPOMI+IASI data set. Figures (see 57$^{th}$ comment) and corresponding results are updated accordingly.

12. Line 213 ff: "To remove this influence, we calculate the tropospheric CAMS forecasts CH$_4$ (TXCH$_4$) from the surface up to 7 km." Why 7 km? The height of the tropopause surely changes over the course of the three-year observation period.

The following statement has been added to the paper:

"XCH$_4$ is affected by local surface emissions and a varying stratospheric contribution due to changes in the tropopause altitude (Liu et al, 2021; Schneider et al., 2021). This stratospheric contribution has to be taken into account, in order to be able to use XCH$_4$ for a reliable investigation of local surface CH$_4$ sources and sinks (Pandey et al., 2016). Our background removal method effectively accounts for the stratospheric contribution. To show this we apply the approach to CAMS forecasts of XCH$_4$ (which has a significant stratospheric contribution) and TXCH$_4$ (calculated from the CAMS forecast as the CH$_4$ averaged from surface to 7 km, which should have a very limited stratospheric contribution). The results are presented in Figure 5d-f. The CAMS TXCH$_4$ anomalies have similar distribution as CAMS XCH$_4$ anomalies, suggesting that our background removal approach reliably removes the stratospheric contribution."

13. Line 233-234: „In addition, the downwind plume is similar to the cone shaped plume in our simple plume model ..." What do you mean by similar? Just the spatial occurrence? As you use three different colormaps it is hard to judge by eye. It is clear, that the modelled cone-plumes result in XCH$_4$ enhancements which are smaller by a factor of two or even more, suggesting that the CAMS-GLOB-ANT emissions are too low.

Thanks, we have used the same colormaps for the first two figures (see 16$^{th}$ comment).

It is true that the XCH$_4$ enhancements derived from the modeled cone plumes are lower. This cone-plume model only considers a simple linear proportion of wind speed and emission strength. Huge

biases are expected in a simple day or in a short period. But these biases can be compensated over a long-term period.

The three kinds of XCH₄ enhancements on the example day have similar spatial patterns, which help to support the reasonable assumption of a cone-shape distribution. The corresponding sentence in the text has been changed.

14. Line 234: "... which implies our model assumption is reasonable." Either CAMS-GLOB-ANT has too small emissions, or the model generates a systematic bias. See comment above.

The sentence has been changed to:

"In addition, the spatial pattern of the downwind plume is similar to that of the cone-shaped plume, which implies our cone-shape assumption is reasonable."

15. Figure 5 correlation plots (c & f): I assume the gray line is the bisector. Please include your regression line. Please do so also for the other correlation plots in the manuscript.

The figures have been updated, see the 57th comment.

16. Figure 6: colormaps: Why a diverging colormap for wind speed? Is 4 m/s a representative mean value? If so, please indicate this in the caption, otherwise I would suggest a perceptually uniform sequential colormap. Also, please do not use the same colormaps for windspeed as for XCH₄ enhancements or wind-assigned anomalies.

Thanks, the figure and its caption have been updated.

[Figure]

Figure 6: ΔXCH₄ together with the ERA5 wind at 12:00 UTC from (a): TROPOMI observations at 11:34 UTC, (b): CAMS forecast at 12:00 UTC, and (c): from the simple plume model (averaged over the daytime) based on the CAMS-GLOB-ANT inventory over the USCB region on an example day (6 June 2018). The "bg" in the title of (a) and (b) represents the background, derived from the mean XCH₄ in the upwind region (50.3º-50.8º N, 19.5º -20.0º E).

17. Figure 6 colormaps: Why is TROPOMI transparent/shaded and the other two are not? Please have consistent colormaps for all plots. Especially the modeled plume has much smaller values than CAMS and TROPOMI. This becomes more difficult to see with the currently used colormaps.

The figures have been updated (see the 16th comment).

18. Figure 6: Do I understand correctly that CAMS forecast is at 12 UTC? Is TROPOMI also at 12 UTC? Is the modeled plume an average over 2018-06-06 or also at 12 UTC? Please clarify.

The CAMS forecast is at 12 UTC. The TROPOMI observation is around 11:34 UTC. The modeled plume is an average over 2018-06-06 (8:00 UTC – 18:00 UTC). This information has been added to the caption (see the 16[th] comment).

19. Figure 6: As you are showing snapshots of a specific time you should also give the respective value of the background term that has been subtracted. Either in the caption or the title.

The backgrounds for (a) and (b) have been added to the captions of the subfigures (see the 16[th] comment).

20. Figure 5 & 7: Why are the wind-assigned anomalies far more south-west than the CAMS forecasts/observations? Is the ERA5 wind wrong or is there a methodological error?

The CAMS data include both background and emissions from the sources. Compared to the background, the enhancements are tiny signals. Therefore, these background signals need to be removed before simulating the wind-assigned anomalies.

The CAMS enhancements (Figure5/7-a) contain all wind information (i.e., NE, SW, and rest). The wind-assigned anomalies are the difference between NE and SW. Moreover, there are more emission sources with higher emission rates in the SW direction of the study area. The plume, therefore, tends to cover the SW region. The enhancements for NE and SW wind are shown below:

[Figure]

21. Figure 5 & 7: I assume "CAMS emission (9.7E26 molec./s)" is the sum of all sources in the CAMS-GLOB-ANT inventory and used as an a priori value for the calculation of the wind-assigned anomalies, is this correct? Please clarify in the caption and do not repeat as a title for every single plot. Same for Figure 7.

The referee is right that we use the CAMS-GLOB-ANT inventory as the a priori values and locations of the sources for calculating the wind-assigned anomalies.

Figures have been updated (see the 57[th] comment).

22. Line 253: "The TROPOMI+IASI result has a slightly higher uncertainty than the TROPOMI result". Please remove "slightly". The uncertainty is more than a factor 4 higher.

Changed accordingly. We also removed "slightly" from the sentence on line 344 in the conclusion.

23. Line 263: „... the emission rate uncertainties of using $XCH_4$ or $TXCH_4$ are insignificant compared to the estimated emission rates." Please change "insignificant" to "small" or something equivalent. If uncertainties were insignificant, they would not need to be reported.

Changed accordingly.

24. Line 287: "Considering the height of the Planetary Boundary Layer (PBL), we use the ERA5 wind at 500 m above the ground (Figure 3c)" I don't understand what you mean by that. The PBL height changes during the day and year. How is 500 m related to the PBL height? In your abstract you give the emission rates for a height of 330 m.

The enhancement in $XCH_4$ that is being used to estimate the emission rates is composed of $CH_4$ molecules that have been advected in different heights. To me it is unclear why a certain height should be more representative than the other (at least in the PBL). Shouldn't an average wind speed over the entire vertical spread of the PBL be used. This would of course massively increase the uncertainty of your estimation. Please comment.

The PBL thickness over the study area ranges from 700 m to 1.5 km over the daytime in summer, which is about 1 km on average (Luther et al., 2019; Krautwurst et al., 2021). Winds typically increase above and decrease below the middle of the PBL. Thus, we chose the altitude (~500 m) in the center of PBL for an explicit averaging over the PBL thickness, and wind at 500 m is used to investigate the uncertainty of wind (see the 27th comment).

This sentence has been rephrased to:

"Assuming that the height of the Planetary Boundary Layer (PBL) is typically less than a kilometer, we use the ERA5 wind at 500 m above the ground (Figure 3c) for describing the transport of methane released in the study region."

25. Figure 9: The grids that are shown are a superposition of the days on which, in the daily average, the wind blew in the respective narrow-regimes, right? If so, this should be explained once more in text in section 3.3.2. I don't understand where all the missing data points originate from. If for example the cone-plumes are never advected into the narrow wind-regime at NW-SE, then, for the respective grid, the calculation is 0-0=0, isn't it? If so, you can of course filter these grids to make a distinction to cases where $XCH_{4NW}$ and $XCH_{4SE}$ are equal but not zero. If this is being done please explain it somewhere in the text.

If a wind direction dominates 60% of records for one day, i.e., if the wind direction belongs to one specific area ($NW_{1/2}/SE_{1/2}$) more than 60 % of the daytime (08:00–18:00 UTC), then this predominant wind direction is selected for that day. When narrow wind regimes are used, the number of days with $NW_{1/2}$ (or $SE_{1/2}$) wind is much less than that with NW wind, i.e., fewer data points for $NW_{1/2}$ (or $SE_{1/2}$) regimes. Moreover, to eliminate the biases, we select the grids with more than 10 measurements. The fewer data points result in more missing grids here.

26. Line295 ff: „The final estimated emission strength is weighted by the number of the valid binning data in the plume maps under different wind regimes (i.e. 171 for narrow NE-SW and 26 for narrow NW-

SE, respectively)." I do not understand the weighting. Are there 171+26 days in total? The emission rate of 9.8E26 molec./s from NE-SW regime is being weighted with 171 days and the 14.0E26 molec/s with 26 days. Result is then 10E26 molec/s which is given in line 303? If I understand correctly please insert the information that by "number of the valid binning data" you mean "number of days on which, on average, the wind blew in the respective wind-regime."

We used the number of valid grids in the wind-assigned anomalies as the weighting, but this might be not fully accurate. The total days for $NE_{1/2} - SW_{1/2}$ (115 days) and for $NW_{1/2} - SE_{1/2}$ (71 days) should be used as the referee recommended.

The sentence has been rephrased as follows:

"The final estimated emission strength is weighted by the number of days on which, on average, the wind blew in the respective wind regime (i.e., 115 days for $NE_{1/2} - SW_{1/2}$ and 71 days for $NW_{1/2} - SE_{1/2}$, respectively)."

Since the CAMS data are collocated to the TROPOMI+IASI TXCH4 (see the 11[th] comment), the corresponding results in the paper are also changed:

"The estimated emission rate is about $773 \pm 13$ kt/year ($9.2E26 \pm 1.6E25$ molec./s) for the $NE_{1/2} - SW_{1/2}$ field. This indicates that the effect of the segment in the wind field coverage is negligible when there are enough measurements. The use of $NW_{1/2} - SE_{1/2}$ wind fields yields an emission strength of $1176 \pm 109$ kt/year ($1.4E27 \pm 1.3E26$ molec./s). The higher uncertainty is probably due to less measurements in these wind fields. The weighted rate is therefore about 927 kt/year (1.0E27 molec./s), 13.4% higher than based on the wider NE-SW wind regime (Sec. 3.1)."

27. (1) Line 352 ff: "However, their speeds decrease by 19% at 10 m and increase by 32% at 500 m, which results in higher emission rates by -23% and 13 %, respectively." How can that be? Wind-speed is linear in the calculation of ε, isn't it? Accordingly, the emission rates should also be -19% & +32%. Please comment.

(2) Furthermore, "higher emission rates" is not correct for describing a decrease and an increase. Please rephrase.

(1) We used TROPOMI data over a larger area before applying the coincidence criteria with the wind. Since we decided to collocate CAMS forecast data to the TROPOMI+IASI data set (see the 11[th] comment), the wind information is slightly changed (see Table 1 below). The weighted-average wind speed at each level is calculated based on the days at each wind regime. The wind speed reduces by 20% at 10 m, and increases by 32% at 500 m, compared to that at 330 m. The corresponding changes in the estimated emission strengths are -25% and 13%, respectively. These values and Table 1 have been updated in the paper.

Table 1: Number of days and the averaged wind speed (± standard deviation) per specific wind area in daytime (08:00 UTC – 18:00 UTC) at different vertical levels from November 2017 to December 2020 over the USCB region. The days for the three-year average coincide with the TROPOMI overpass days.

| | NE / >315° or <135° | | SW / 135° – 315° | |
|---|---|---|---|---|
| | Number of days in total (%) | Averaged wind speed ± standard deviation (m s⁻¹) | Number of days in total (%) | Averaged wind speed ± standard deviation (m s⁻¹) |
| **10 m** | 39.1 | $3.2 \pm 1.5$ | 56. 9 | $3.4 \pm 1.6$ |
| **~330 m (975 hPa)** | 38.7 | $4.1 \pm 2.2$ | 56.9 | $4.3 \pm 2.3$ |
| **~500 m (950 hPa)** | 38.7 | $5.0 \pm 2.7$ | 57.3 | $5.9 \pm 3.5$ |

"The wind speed is linear in the calculation of $\varepsilon$, but the wind speeds do not all linearly change for each grid and for each time at different levels. This results in unequal changes between the wind speed and the enhanced columns, and later unequal changes in the estimated emission strength. In addition, the simple cone plume model introduces biases, i.e., the enhanced column in the downwind is set to zero when its location is out of the cone angle (60º). Slight changes in the wind directions might result in a huge difference in the enhanced columns." This statement has been added to the text.

The figure below shows the correlation plots for the enhanced columns at 10 m and 500 m, compared to the ones at 330 m. The changes in the enhanced columns are -26% and 14%, which are similar to the changes in the estimated strengths.

[Figure]

(2) the sentence has been changed to:

"However, their speeds decrease by 20% at 10 m and increase by 32% at 500 m, which results in changes in the emission rates by -25% and 13 %, respectively."

**Technical comments**

28. Please consider perceptually uniform sequential colormaps, especially for figures 2, 4 and 6. Diverging colormaps are helpful in displaying differences, which in your case would only make them suitable for plotting wind-assigned anomalies and XCH₄ enhancements. If you stick to the red-blue diverging colormap for the anomalies consider hatching the grids with missing data. At the moment they are easily mistaken for value 0.

   If you are using python to generate plots you might have a look here:

   https://matplotlib.org/2.0.2/examples/color/colormaps_reference.html

   Thanks, your comment is helpful for improving the figures. We modified the figures and in the appendix we added a zoom into the area shown in Figure 4(b).

[Figure]

Figure 1: Spatial distribution of (a) the CAMS global anthropogenic emissions from all sectors and (b) percentage share of the fugitive emissions compared to the overall anthropogenic emissions over the USCB region on a 0.1° × 0.1° latitude/longitude grid. The fugitives are the dominant CH₄ sources.

[Figure]

Figure 4: averaged (a) CAMS forecast XCH₄, (b) TROPOMI XCH₄ and (c) TROPOMI+IASI TXCH₄ in the USCB region on a 0.1° × 0.1° latitude/longitude grid during November 2017-December 2020. The square and triangle symbols represent the locations of CAMS-GLOB-ANT sources (for a better viewing, only the emission strengths larger than 1E24 molec./s are shown here) and the active coal mine shafts from the CoMet inventory (Gałkowski et al., 2021), respectively. Different colors denote the amount of emission rates. The white grids represent no data from TROPOMI or the number of the points in the grid less than 5. A zoom version of panel (b) is shown in the appendix (Figure A- 2Figure A- 2: A zoomed version of Figure 4(b).). Note, a different colorbar has been used in panel (c).

[Figure]

Figure A- 2: A zoomed version of Figure 4(b).

For Figure 6, see the 16th comment.

29. For many citations there is a dot missing after "et al".

Thanks, changed accordingly.

30. For almost all figures the labeling is way too small. Please increase the font size corresponding to the text.

Thanks, the figures have been updated.

31. Line 12: „Intensive coal mining activities are in the Upper Silesian Coal Basin (USCB) in southern Poland, resulting in large amounts of methane ($CH_4$) emissions." Maybe shift the "are" in front of "resulting".

Changed accordingly.

32. Line 13: "Annual $CH_4$ emission reached to 448 kt according to the European Pollutant Release and Transfer Register (E-PRTR, 2017)." Please remove the "to" or change to "... reached up to 448 kt ..."

Changed accordingly.

33. Line 14-15: "As a $CH_4$ emission hot spot in Europe, it is of importance to investigate its emission sources and accurate emission estimates". Maybe insert "make" in front of "accurate emission estimates"

Changed accordingly.

34. Line 16: "column-averaged dry-air molar fraction observations of $CH_4$". Please change to "mole fraction observations".

Changed accordingly.

35. Line 16-20: It is a rather long sentence. Maybe split up.

The sentence is split up into two sentences as the referee recommended:

"In this study, we use satellite-based column-averaged dry-air molar fraction observations of $CH_4$ ($XCH_4$) from the TROPOspheric Monitoring Instrument (TROPOMI) and tropospheric $XCH_4$ ($TXCH_4$) from the Infrared Atmospheric Sounding Interferometer (IASI). In addition, the highresolution model forecast $XCH_4$ and $TXCH_4$ from the Copernicus Atmosphere Monitoring Service (CAMS) are used to estimate the $CH_4$ emission rate averaged over three years (November 2017 to December 2020) in the USCB region (49.3° - 50.8° N and 18° - 20° E).”

36. Line 27 ff: “... with using the Carbon dioxide and Methane (CoMet) inventory ...” What information is actually used from the CoMet inventory? As you report your emission estimates in the next sentence I assume that, here, you just take the locations of the shafts. Please be more specific, as the CoMet inventory also reports emission rates of individual shafts.

We used both the location and the proportion of the emission rate for each shaft in the total emissions, so as for the CAMS-GLOB-ANT. This information is added:

“Using the CAMS inventory (CAMS-GLOB-ANT) as the a priori knowledge (location and the proportion of the emission rate for each source in the total emissions) of the sources, together with ERA5 wind at 330 m, the wind-assigned $XCH_4$ anomalies for two opposite wind directions are calculated, which yields an estimated $CH_4$ emission of 815 kt/year ($9.7E26 \pm 1.5E25$ molec./s) for CAMS $XCH_4$ and 798 kt/year ($9.5E26 \pm 1.3E25$ molec./s) for CAMS $TXCH_4$.”

37. Line 28: Not sure what is meant by “performed”? An inventory is not performed. How about “... from 2018”, “... covering the year 2018”, “issuing the year 2018” or something equivalent.

Thanks, the sentence has been changed to:

“This wind-assigned method is further applied to the TROPOMI $XCH_4$ and TROPOMI+IASI $TXCH_4$ with using data from the Carbon dioxide and Methane (CoMet) inventory derived for the year 2018.”

38. Line 34-35: “When using different wind coverage and different wind segmentation, an uncertainty of 4.2% and -2.1% is obtained, respectively”. How is an uncertainty negative? Maybe uncertainty is not the adequate word.

The referee is right that this word is not properly used. The “uncertainty” here meant the changes in emission strength when different wind information was used. The sentence has now been modified to:

“When using different wind coverage and different wind segmentation, the estimated emission strengths change by 4.2% and -2.1%, respectively.”

39. Line 40-42: This sentence is hard to read. In my opinion the word “and” is used to often. I think in “... and waste disposal ...” you can remove it.

The sentence has been changed as the referee recommended:

“Methane sources induced by anthropogenic activities include fossil fuel production and use (e.g., coal mining, gas/oil extraction), waste disposal, and agriculture, which in total accounts for about 60% of the total $CH_4$ emissions (Saunois et al., 2020).”

40. Line 43: „... to the atmosphere $CH_4$ level are still ...“. This seems off. Maybe change to “atmospheric” or “atmosphere’s”

Changed accordingly.

41. Line 75: "... data sets provide a large coverage and long-term $XCH_4/TXCH_4$ observations, which helps to better estimate $CH_4$ emission ..." I guess it should be "help", not "helps".

Changed accordingly.

42. Eq. 1: The square root should also include the numerator. The calculation of the standard deviation is trivial. If you want you can remove the equation.

The equation has been removed as the referee recommended.

43. Line 111: "... emissions from ships with a magnitude of 19 are much lower ...". What do you mean by "magnitude"? Do you mean the "count" of ships?

It should be the "orders of magnitude of the emissions". The sentence has been changed to:

"The emissions from the sectors "agriculture soils" and "solvents" are zeros. The $CH_4$ emitted from ships has 19 orders of magnitude, which are much lower than the other sectors".

44. Line 113: "Compared to its high amount, the seasonal variations of the fugitives sector can be ignored." Sounds off to me. Maybe avoid "high amount" when referring to emission rates. I suggest "The seasonal emission variations of the fugitive sector are minor and can be ignored" or similar.

The sentence has been changed as the referee recommended:

"The seasonal variations of the fugitives sector are minor and can be ignored."

45. Line 113 – 119: From here on, the paragraph is no longer stringent to me. What the authors are basically saying is that the fugitive sector is dominant in the USCB. As the fugitive sector has minor seasonal variations they do not consider them. I suggest the following restructuring from line 111 onwards: "Thus, these three sectors are not shown here. The sources from agriculture livestock ($1.7E25 \pm 4.0E25$ molec./s) amount only 4% of the total emissions in this region. The dominant $CH_4$ sources in this region are fugitive sources from from energy production and distribution (e.g. fuel use). With a mean value of $7.9E26$ molec./s and a standard deviation of $2.2E25$ molec./s they account for 82% of the anthropogenic $CH_4$ emissions in the CAMS-GLOB-ANT inventory ($9.7E26$ molec./s in total). This becomes particular visible in the spatially overlapping distribution within the USCB (see Figure 2). The seasonal emission variations of the fugitive sector are minor and can be ignored. Therefore, we apply the three-year mean of total emissions at grids with significant emissions without considering seasonal variations in the simple plume model (see Sect. 2.3)"

We would like to thank the referee for rephrasing the sentences. The sentences are changed as the referee recommended.

46. Figure 1: The coloring is highly unfortunate. In the legend the "Fugitives" is listed last and easily mistaken for "Off road transportation". Please list "Fugitives" first and change the color for "Off road transportation".

The figure is updated to:

[Figure]

Figure 2: Stacked area plot for different sectors of the monthly averaged CAMS global anthropogenic emissions (>1E20 molec./s) in the USCB region for 2017-2020 (https://permalink.aeris-data.fr/CAMS-GLOB-ANT, last access: 22 December 2021. Granier et al., 2019).

47. Figure 2: I assume that the barely visible gray lines are the borders to the Czech Republic and Slovakia. Please increase the resolution of the basemap so that the borders can be recognized as such. For a better orientation you might consider inserting country abbreviations.

At first glance, the two heatmaps look identical, which is of course the point being made here. However, I'm a bit unsure about the gain of information when two nearly identical images are shown side by side. Perhaps a heatmap of the percentage shares of fugitive emissions compared to overall anthr. emissions would be better. Please comment.

Thanks, the figure in terms of percentage share helps readers to better understand that fugitive emissions are the dominant sources. The figures have been updated (see the 28[th] comment)

48. Line 146: Comma before "which"

Changed accordingly.

49. Line 147: "... it is able ..." What does "it" refer to? I guess it refers to the combined product, which you introduce as such only in the following sentence. I suggest "... we are able to ...".

Changed accordingly.

50. Line 154: You reference the figure 2 from a previous publication. In my opinion it would be beneficial to actually show the figure again.

The figure has been added in the Appendix.

[Figure]

Figure A- 1: Sketch of the simple plume model used to explain the $CH_4$ emission estimation method. The methane at the point source is distributed along the wind direction (wind speed: $\boldsymbol{v}$) in the cone-shaped area with an opening angle of $\alpha$. The point source emits the methane at an emission rate of $\varepsilon$. We assumed the methane molecules are evenly distributed in the dotted area A, and the distance from area A to the point source is d. Therefore, the emitted methane in dt time period equals to the amount of methane in the area A. It yields the equation $\varepsilon \times dt \approx \Delta column \times \frac{\alpha}{\pi} \times \pi \times d \times v \times dt$. This figure is adopted from Tu et al. (2022).

51. Line 159 and Eq. 2: The indices i of (xi, yi) should be subscripted.

Changed accordingly.

52. Figure 3: The individual plots in Figure 3 will separately be referred to with a, b and c (e.g. Line 288). Please add a numbering to the plot or change reference in the text to left, middle and right.

Changed accordingly.

53. Figure 4: Please increase the size of the squares for the CAMS-GLOB-ANT sources. The color of the sources is so difficult to distinguish. An increase in the size of triangles for the CoMet sources would also be beneficial, although this might be more difficult as the triangles overlap. If possible, please improve the visibility. If not, you might consider providing a zoom to the shafts in a separate subsection, which was suggested by Referee#1.

Changed accordingly. See the 28th comment.

54. Figure 4: In the caption it says "during November 2017-December 2020". Does this mean the displayed $XCH_4$/$TXCH_4$ data are an average of this period? If so, please indicate this in the caption. Otherwise, please specify the displayed day.

The data in the figure represent the average. The information has been added in the caption as the referee recommended (see the 28th comment).

55. Figure 4: I assume that the white grids are missing data. Please indicate this in the caption. Moreover, the color choice is unfortunate, as it is missing values are difficult to distinguish from the mid-range values in the colorbar. Please see my earlier comment regarding colormaps.

The figure is updated (see the 28th comment).

The sentence has been added in the caption:

"The white grids represent no data from TROPOMI or the number of the points in the grid is less than 10."

56. Figure 4: For a better comparison please consider using an identical colormap for a) and b). The TROPOMI & IASI data product has of course higher values. If the colorbar consists of the same colors, please indicate the shift in values in the caption.

The figure has been updated (see the 28th comment).

57. Figure 5 & 7: If a diverging colormap is being used, please center the colorbar to the value 0. Please use the same colormap for all four plots.

[Figure]

Figure 5: (a)-(c): CAMS XCH$_4$ enhancement anomalies (XCH$_4$-background), the wind-assigned anomalies (NE-SW), and correlation plot of the wind-assigned anomalies between CAMS and the cone plume model with using the CAMS-GLOB-ANT inventory (9.7E26 molec./s in total) and ERA5 wind at 330 m during November 2017-December 2020 over the USCB region. (d)-(f): the same as for the upper panel but for CAMS TXCH$_4$ (colorbars in (d) and (e) are different from that for XCH$_4$). The square symbols represent the locations of the CAMS-GLOB-ANT (>1E24 molec./s) inventory and different colors denote the amount of emission rates. The hatched areas in (a)-(b) and (d)-(e) represent no data in these grids. The uncertainties in (c) and (f) represent the mean error bars, i.e., error propagation of the background uncertainty and the CAMS standard deviation.

[Figure]

Figure 7: Similar to Figure 5, but for (a-c) TROPOMI $XCH_4$ and (d-f) TROPOMI+IASI $TXCH_4$. The a priori knowledge of sources are based on the CoMet inventory (Gałkowski et al., 2021). The triangle symbols represent the locations of the active coal mine shafts and different colors denote the amount of emission rates.

58. Figure 5 & 7: Please avoid the term "anomalies" if you are not referring to wind-assigned anomalies. Rather use "enhancement" as suggested in an earlier comment

Changed accordingly. See the 57th comment.

59. Figure 5 & 7: Please do not repeat the identical title for multiple plots in the figure. I Suggest to name the lines on the left with [$XCH_4$, $TXCH_4$].T. Name the columns with [CAMS, modelled (cone-plumes + ERA5), correlation plot]. Instead of "modelled (cone-plumes + ERA5)" you could of course choose a term of your choice. Something like "wind-assigned anomalies (SW- NE)" or similar would be fine too.

Changed accordingly. See the 57th comment.

60. Figure 5 & 7: The colorbar-label for the left plots (a & d) and the middle plots (b & e) are currently the same. The left plots are displaying $XCH_4$ enhancements (i.e. $XCH_4$ – background), the middle plots are displaying wind-assigned anomalies. Please correct the colorbar labels.

Changed accordingly. See the 57th comment.

61. Figure 5 & 7 caption and title of middle plots: "... the wind-assigned anomalies (NE-SW) ..." Shouldn't it be "SW-NE"? Otherwise the positive values should be in the NE.

The NE wind results in higher values in SW region, vice versa. Therefore, NE-SW results in positive values in SW region. See the figures in 20th comment.

62. Figure 5 & 7 correlation plots (c & f): Please remove the title. The information is already given in the axis' labels. Also, as mentioned before, the use of $\Delta XCH_4$ is not consistent.

Changed accordingly. See the 57th comment.

63. Line 216: "9.1E24 ± 1.2E24 molec./s" I guess there is a typo in the exponent.

Thanks, it should be 9.1E26 ± 1.2E24 molec./s (i.e., 798 ± 11 kt/year). Changed accordingly.

64. Line 227: "Figure 6 illustrates the enhance $XCH_4$ (raw $XCH_4$-background in the upwind) distribution ..." Please correct "enhance" to either "enhanced $XCH_4$" or "$XCH_4$ enhancement". Why is "in the upwind" specified? From the explanation in the appendix of your earlier publication the background determination is not limited to the upwind.

The "enhance" has been changed to "enhanced".

The referee is right that we determine the background is not limited to the upwind, but based on long-term observations, from which the seasonal cycle, linear increase, etc. are removed. Here, the enhancements from three different datasets on a single day are shown as examples, and thus, we use the $XCH_4$ in the upwind as the background.

65. Line 244: "... anomalies show high amounts around the areas ..." To me "amounts" sounds off. Please consider something like "high concentrations", "high methane content" or something similar.

The "high concentrations" is used, as recommended by the referee.

66. Figure 8: Please be precise in the labeling of the horizontal lines, i.e. "Total Emission (CAMS- GLOB-ANT", "Total Emission (CoMet inventory)".

We would like to thank the referee for these comments to improve the figure. Changed accordingly.

[Figure]

67. Figure 8: Please remove the shaded background and instead add a legend: "a priori: squares CAMS-GLOB-ANT, triangles CoMet inventory", or something similar. If plotted among each other triangles and squares are easier to compare.

Changed accordingly. See the 66th comment.

68. Figure 8: The error bars are very small, as you mention in the caption. Nevertheless, please use either a uniform color, such as black or gray, or simply the color of the respective marker. At the moment it seems like they change colors randomly.

Changed accordingly. See the 66th comment.

69. Line 280: „Here we investigate the wind uncertainties ...". Please insert a comma after "here"

Changed accordingly.

70. Line 284: "Compared to the wind at 330 m, the wind distributions are similar ..." Please specify, in the whole text, that you are referring to the distribution of wind directions.

Changed accordingly.

71. Section 3.3.2: Since the designation SW and NE were used previously and now SW and NE are still used for narrow, the text is a bit confusing. Either _narrow is always subscripted consequently, as is being done in the caption of Fig. 9, or, alternatively, the subscripts SW1/2 or SW1/4 could be used to specify whether the wind field is divided into halves or quarters.

Using the subscripts as the referee recommended, is a better way to make the text clearer. We use NE$_{1/2}$ for 0°-90°, SW$_{1/2}$ for 180°-270°, NW$_{1/2}$ for 270°-360°, and SE$_{1/2}$ for 90°-180°.

72. Figure 9 and text in section 3.3.2: Isn't it "SW-NE" instead of "NE-SW"?

It should be NE-SW (see the 61$^{st}$ comment).

73. Line 310: "The wind category here is based on its predominant wind fields over the USCB region ...". Please change "its" to "the" or rephrase.

The sentence has been changed as the referee recommended:

"The wind category here is based on the predominant wind fields over the USCB region and is divided into two opposite sectors (SW and NE)."

74. Line 311: "To investigate its uncertainty, we apply another kind of segmentation:" What does the "its" refer to? Please change to "To investigate the effect of the segmentation on the uncertainty in the emission rate estimation, we additionally apply another kind of segmentation" or similar.

The sentence has been changed as the referee recommended:

"To investigate the effect of the segmentation on the uncertainty in the emission rate estimation, we additionally apply another kind of segmentation: N (<90° or >270°) and S (90° - 270°) categories."

75. Line 335: "To investigate the CH$_4$ emissions from this hot spot, the CoMet campaign was performed in 2018. Locations and emission rates of the ventilation shafts of the coal mine used in this study are based on this inventory". "This" probably refers to the CoMet campaign. A campaign is not an inventory. Please rephrase.

The second sentence has been changed as the referee recommended:

"Locations and emission rates of the ventilation shafts of the coal mine used in this study are based on this campaign."

76. Line 340: "... and reasonablely compared to the CoMet inventory (6.6E26 molec./s)" Please change "reasonablely" to "reasonable"

Changed accordingly.

77. Line 343: "... up to 5.68E26 molec./s derived from one flight (Kostinek et al.(2021)). Similar 2D anomalies and plumes are also observed ..." Similar to Kostinek et al.? Otherwise, please separate into two paragraphs to make it clear that you are now writing about plumes/anomalies and no longer about total emission estimates.

Changed accordingly.

78. Table A-2: Instead of "CAMS emission" & "shafts emission" I think it would be better to use " CAMS-GLOB-ANT" and "CoMet inventory" according to the caption. In the left column you could label the line as "prior emission sources" or similar.

Thanks, changed accordingly.

79. WMO Reference from Line 40 is missing.

The reference has been added.

---

## Referee Report (RR1)

**General comments:**

The authors have made an impressive effort to improve the manuscript. The visual appearance of all figures has improved significantly. The text also shows significant improvements. Many stumbling blocks for the reader have been removed.

Nevertheless, some comments remain incompletely addressed. In particular, there are individual comments which the authors claim to have incorporated, but did not do so completely in the manuscript or did so incorrectly.

Concerning the improved Figures, I have to apologize. I've noticed that in my previous comments I've sometimes mixed up the terms "colormap" and "colorbar". At the current state I wonder why for some figures the colorbars cover different values. In general, it is obvious that $TXCH_4$ shows stronger enhancements than $XCH_4$. But for example, Figures 5 (a) and (b) should have the same range of values. (d) and (e) should also cover the same values. Otherwise, the visual comparison of the plots side by side creates a false impression. The same holds true for Figure 7, Figure A-2 and Figure A-3. If there is a reason to not use that same range of values in the colorbars, please indicate so in the respective captions. Like you did in Figure 6, where you're only want to compare spatial pattern.

Comments of Referee #1:

- 1.1: I strongly agree with Referee #1. The CAMS data was used solely for the validation of the wind-assigned anomaly method. The term "using" suggests that the CAMS data was actually used for your emission estimates, which they where not. I would even go one step further than Referee #1 and not include the CAMS data in the title at all: "Quantifying hard coal mines $CH_4$ emissions from TROPOMI and IASI observations using the wind-assigned anomaly method". However, this is only a recommendation. If the authors feel differently I suggest discussing it with the editor.

- 2.1: for the sake of completeness, a study on measurements on HALO should also be cited, as HALO was the flagship of the campaign. I recommend Galkowski et al. 2021 (https://doi.org/10.5194/amt-14-1525-2021) on in situ observations on HALO and Wolff et al. 2021 (https://doi.org/10.5194/amt-14-2717-2021) on airborne lidar observations. Also, I want to raise your attention to Andersen et al. 2021 (https://doi.org/10.1016/j.aeaoa.2021.100135) on UAV based emission estimates in the USCB and Luther et al. 2022 https://doi.org/10.5194/acp-22-5859-2022. If it seems fitting to you, you could include these two publications at an appropriate location of your manuscript. But of course, only as an option for you.

**Specific comments:**

- SC 0 (new comment):
  - In the abstract in lines 21-27 validation results are given. I'm in big favor of giving results in the abstract, but only the main results, i.e. the emission estimates based on the satellite observations. Here, it is sufficient to simply state, that the wind-assigned anomaly method is validated using CAMS forecast data, showing good agreement to the CAMS-GLOB-ANT inventory. You don't have to give numbers. For the reader the results of the validation distract from the main results, which are supposed to be the highlight in the abstract.
  - The same applies for the last paragraph of the abstract (i.e. lines 37-43). The sensitivity analysis of wind speed is a method for determining a contribution to the uncertainty in the emission estimates. The results of this analysis should be reflected in the given uncertainty of the emission estimates. For the abstract it is sufficient to state that a sensitivity analysis of wind speed for different altitudes has been made.

This is part of your chosen approach and should be stated before giving the main results.

- SC 3: In line 201 it now says "wind regime sector". In Table 1 it says "wind area". Please recheck the manuscript, if all your changes are applied to your will.
- SC 4: It still says "simple plume model" in the title of subsection 2.3, in lines 131, 185, 277, 400, captions of Fig. 6 and A-1. Please review your entire manuscript for consistency and use only "cone plume model" or "simple cone plume model".
- SC 5: If there are such high spreads and uncertainties in your estimates for the individual years, I don't understand how you come up with such low uncertainties in your estimate of the three-year period. Please comment this in the scope of SC 24/27 below.
- SC 6: Eq. 6. I'm a bit baffled by the mix of equation and free text. I have to admit that I'm not sure if that's formally allowed or not. You might think of a variable for "wind-assigned anomaly". Something like "$\delta XCH_4$" or similar. I realize that you would only need this variable at this point so this is only a recommendation. You could also wait for a comment from the type setting of the journal.
- SC 12: I'm afraid you didn't add your statement to the manuscript. At least in the "track-changed" it is unchanged (see 256 ff). Moreover, you still haven't answered the question. How did you come up with exactly 7 km? Why not 6 or 8 km?
- SC 13/14: "This cone plume model only considers a simple linear proportion of wind speed and emission strength. Huge biases are expected in a simple day or in a short period. But these biases can be compensated over a long-term period." If you do not explain how these biases come about, it is hard to understand why they are compensated over a long period. I do not understand why you expect such huge biases. Most plume models (e.g. Gaussian plume model) are recursively linear with wind speed and linear with emissions rate. Usually, in plume models some parameter accounts for turbulence. The only possible representation of turbulence in your model is in the angle α=60°. In some cases, this angle will be too small, in some cases it will be too large. This effect might be canceled out, as you suggested, over the long observation period. But it seems as your cone-model is either showing lower $XCH_4$ enhancements, or supposedly too narrow plumes. As your overall goal is to be representing for the overall observation period this is no show-stopper, but please openly discuss/explain the limits of the cone plume model to the reader.
In your former publication you showed plots of the cone plumes for different values of α. While you derived 60° as the best fit to the $N_2O$ plumes for Madrid, I'm not convinced that 60° is necessarily the optimal fit for the USCB, too. At the very least should investigate to what quantitative extent variations in α impose uncertainties on your emission estimate. As far as I can see this has not yet been considered in your uncertainty analysis Sect. 3.3.
- SC 16 Figure 6: Why do the colorbars start with the value 5 and not 0? If the colorbars are extended (i.e. $\Delta XCH_4<5$ is the same color as $\Delta XCH_4=5$), an extension arrow at the colorbar should indicate so. The same applies for the upper end of the colorbar. But, as mentioned in the general comment above, I strongly recommend using the same range of values for all colorbars. If the spatial pattern in (c) would not be recognizable anymore you can leave the colorbar as it is now., but at least start your colorbars in (a) and (b) with 0.
Also indicate the different colorbars in the caption. Currently it seems like the modelled plumes fit perfectly to the CAMS forecast and the TROPOMI data. Which they do only in spatial appearance, not concerning the magnitude of XCH4 enhancement.
- SC 17: Sorry, in my first review I mixed the terms „colormap" and „colorbars" by mistake. Here, I was actually referring to the colorbars and the different range of values covered by it. As mentioned in SC 16 this is ok, if you mention in the text why you did so.
- SC 20: I actually think that these two plots are of high explanatory value. Please consider including them in the manuscript (optionally). Especially, because I now realized that I

misunderstood, that by NE/SW in the title of your figures you mean wind coming from NE/SW and not plumes propagating in NE/SW-direction. My bad, but if you want to make sure that this will not be misunderstood by the reader, you could include these two plots in the manuscript.

- SC 24/27: In the abstract you give your emission estimate with 479±4 kt/yr and 437±18 kt/year. Then you share the results from your sensitivity analysis regarding wind speed, separately.
  The uncertainties given your sensitivity analysis should be included in the uncertainties of your emission estimates. The $CH_4$ exhaust from the ventilation shafts is released at a height of approx. 20 m. Propagating downwind it will be carried upwards by convective eddies and thereby distributed in the entire boundary layer. Have a look into the video supplement of https://amt.copernicus.org/articles/14/2717/2021/#section11 or the Figure A2. While the highest concentrations of $CH_4$ will, for sure, be advected in the middle of boundary layer, or even closer to the ground, you'll need to include the vertical variations of wind speed in your emission estimate uncertainties in some way!
  In your uncertainty analysis (Sect. 3.3) you analyze three sources of uncertainty. Within the analysis the emission results vary strongly from the validation emission estimates given in the abstract and conclusion. Considering this, I'm confused how the uncertainty in your overall emission estimate can be so small. Please state the relative contributions of the sources of uncertainty to the overall uncertainty, including uncertainties induced by the selection of cone opening angle α (see SC 13/14)

**Technical comments:**

- TC 50: In the caption it still says "simple plume model" although the authors have confirmed to switch to the term "plume cone model". Please watch out for consistency.
- TC 51: In Eq. 2 the indices "i" are still not subscripted. Actually, d and $\Delta CH_4$ are functions of the location. So $x_i$, $y_i$ should not be subscripted, only the index "i":

$$\Delta CH_4(x_i, y_i) = \frac{\epsilon}{v \cdot d(x_i, y_i) \cdot \alpha}$$

- TC 57: caption Fig. 5: "colorbars in (d) and (e) are different from that for $XCH_4$". Actually, all four colorbars cover different values. To me it makes sense to have different colorbars for $XCH_4$ and $TXCH_4$, as $TXCH_4$ generally shows higher enhancements. But why different colorbars for (a) & (b)? And why different colorbars for (d) & (e)? See general comment.
- TC 67: My apologies, by my phrasing "... among each other ..." it was not clear what I actually meant. I thought of something like that:

[Figure]

So that triangles are actually vertically aligned with the squares. In this plot the reader should become aware that CAMS-GLOB-ANT emissions are always higher than from the CoMet inventory. By plotting them vertically aligned this becomes more obvious.

- TC 71: "We use $NE_{1/2}$ for 0°-90°, $SW_{1/2}$ for 180°-270°, $NW_{1/2}$ for 270°-360°, and $SE_{1/2}$ for 90°-180°". My recommendation was to use the subscript "1/2" everywhere when the wind field is divided into two halves (i.e. everywhere before Sect. 3.3.2). The subscript "1/4" was supposed to be used, when the field is divided into quarters (previously designated by "_narrow"). This indexing is of course only necessary when talking about the narrowed angular wind regimes in Sect. 3.3.2. So, if you don't want to include an index in the manuscript before this Sect. it's fine by me. But in Sect. 3.3.2 it should be "1/4".

---

## Author Response (AR2)

**Response to Referee #2**

We would like to thank reviewer #2 again for taking the time to review the revised version of the manuscript. We appreciate the level of detail and the valuable suggestions which help us to further improve the manuscript.

In this author's comment, all the points raised by the reviewer are copied here one by one and shown in blue color, along with the corresponding reply from the authors in black.

**General comments:**

The authors have made an impressive effort to improve the manuscript. The visual appearance of all figures has improved significantly. The text also shows significant improvements. Many stumbling blocks for the reader have been removed.
Nevertheless, some comments remain incompletely addressed. In particular, there are individual comments which the authors claim to have incorporated, but did not do so completely in the manuscript or did so incorrectly.

Concerning the improved Figures, I have to apologize. I've noticed that in my previous comments I've sometimes mixed up the terms "colormap" and "colorbar". At the current state I wonder why for some figures the colorbars cover different values. In general, it is obvious that $TXCH_4$ shows stronger enhancements than $XCH_4$. But for example, Figures 5 (a) and (b) should have the same range of values. (d) and (e) should also cover the same values. Otherwise, the visual comparison of the plots side by side creates a false impression. The same holds true for Figure 7, Figure A-2 and Figure A-3. If there is a reason to not use that same range of values in the colorbars, please indicate so in the respective captions. Like you did in Figure 6, where you're only want to compare spatial pattern.

Thanks for the comment to further improve the figures. Fig. 5 (a) and (d) (and other related figures) represent the overall enhancements (raw $XCH_4$/$TXCH_4$ - background), whereas Fig. 5 (b) and (e) represent the difference between the enhancements for NE wind and for SW wind fields. We think that there is no need to use the diverging colormap in (a) and (d), otherwise it might bring some misleading information here. Thus, a different colormap is used for (a) and (d) (comment from SC 16 is also considered here). For either enhancement figures or wind-assigned anomalies figures, the colorbar covers the same range of values for $XCH_4$ and $TXCH_4$, i.e., (a) and (d) use the same colormap and colorbar, so do (b) and (e).

[Figure]

[Figure]

- 1.1: I strongly agree with Referee #1. The CAMS data was used solely for the validation of the wind-assigned anomaly method. The term "using" suggests that the CAMS data was actually used for your emission estimates, which they where not. I would even go one step further than Referee #1 and not include the CAMS data in the title at all: "Quantifying hard coal mines CH4 emissions from TROPOMI and IASI observations using the wind-assigned anomaly method". However, this is only a recommendation. If the authors feel differently I suggest discussing it with the editor.

Thanks for this suggestion. We agree and have changed the title accordingly.

- 2.1: for the sake of completeness, a study on measurements on HALO should also be cited, as HALO was the flagship of the campaign. I recommend Galkowski et al. 2021 (https://doi.org/10.5194/amt-14-1525-2021) on in situ observations on HALO and Wolff et al. 2021 (https://doi.org/10.5194/amt-14-2717-2021) on airborne lidar observations. Also, I want to raise your attention to Andersen et al. 2021 (https://doi.org/10.1016/j.aeaoa.2021.100135) on UAV based emission estimates in the USCB and Luther et al. 2022 https://doi.org/10.5194/acp-22-5859-2022. If it seems fitting to you, you could include these two publications at an appropriate location of your manuscript. But of course, only as an option for you.

Thanks for recommending these related references.

Andersen et al., 2021 has been already cited in the beginning of the introduction, but its method and result were not introduced. A sentence has been added to the text:

"Active AirCore system aboard an unmanned aerial vehicle (UAV) was used to measure CH4 downwind of a single ventilation shaft and emission rates ranging from 0.5 to 14.5 kt/year based on a mass balance approach and ranging from 1.1 to 9.0 kt/year based on an inverse Gaussian method were estimated (Andersen et al., 2021)."

Luther et al., 2022 was also already cited as an ACPD version in the manuscript ("A recent study (Luther et al., 2021) displays a larger emission rate of 414 – 790 kt/year based on a network of four portable FTS instruments (EM27/SUN) during the CoMet campaign."). We have noticed that the paper has been accepted and thus, we have updated the citation.

The other two studies have been also cited:

"For example, Gałkowski et al. (2021) present results of in situ GHG measurements obtained over nine research flights of the German research Aircraft HALO (High Altitude and LOng Range Research Aircraft) acting as the airborne flagship of the CoMet campaign, together with simultaneous flaks measurements for isotopic composition of $CH_4$. A new lidar CHARM-F ($CO_2$ and $CH_4$ Atmospheric Remote Monitoring Flugzeug) was also onboard HALO and its measurements were investigated to determine $CO_2$ emission rates from the power plant (Wolff et al., 2021)."

**Specific comments:**

- SC 0 (new comment):
  - In the abstract in lines 21-27 validation results are given. I'm in big favor of giving results in the abstract, but only the main results, i.e. the emission estimates based on the satellite observations. Here, it is sufficient to simply state, that the wind-assigned anomaly method is validated using CAMS forecast data, showing good agreement to the CAMS-GLOB-ANT inventory. You don't have to give numbers. For the reader the results of the validation distract from the main results, which are supposed to be the highlight in the abstract.

Thanks for this comment. The abstract related with the CAMS data has been shortened according to referee's comment:

"The wind-assigned anomaly method is first validated using CAMS forecast data ($XCH_4$ and $TXCH_4$), showing a good agreement to the CAMS-GLOB-ANT inventory. It indicates that the wind-assigned method works well."

  - The same applies for the last paragraph of the abstract (i.e. lines 37-43). The sensitivity analysis of wind speed is a method for determining a contribution to the uncertainty in the emission estimates. The results of this analysis should be reflected in the given uncertainty of the emission estimates. For the abstract it is sufficient to state that a sensitivity analysis of wind speed for different altitudes has been made.
  This is part of your chosen approach and should be stated before giving the main results.

The last paragraph is changed to (comments from SC 24/27 are considered as well):

"Uncertainties from different error sources (background removal and noise in the data, vertical wind shear, wind field segmentation, and angle of the emission cone) are approximate14.8% for TROPOMI $XCH_4$ and 11.4% for TROPOMI+IASI $TXCH_4$. These results suggest that our wind-assigned method is quite robust and might also serve as a simple method to estimate $CH_4$ or $CO_2$ emissions for other regions."

- SC 3: In line 201 it now says "wind regime sector". In Table 1 it says "wind area". Please recheck the manuscript, if all your changes are applied to your will.

Changed accordingly.

- SC 4: It still says "simple plume model" in the title of subsection 2.3, in lines 131, 185, 277, 400, captions of Fig. 6 and A-1. Please review your entire manuscript for consistency and use only "cone plume model" or "simple cone plume model".

Thanks. The "simple cone plume model" is used and all the related texts have been changed accordingly. The xlabel of the correlation plots in Fig. 5, 7, 9-11 and Fig. A-4 have been updated as well.

- SC 5: If there are such high spreads and uncertainties in your estimates for the individual years, I don't understand how you come up with such low uncertainties in your estimate of the three-year period. Please comment this in the scope of SC 24/27 below.

The high uncertainties in the first two years mainly come from high uncertainties in the elimination of the background due to the small amount of data. The same situation is met in Sect. 3.3.2 and Fig. 9 (d), when small amounts of data had to be used to derive the estimates for $NW_{1/4}$-$SE_{1/4}$ wind. This high level of uncertainty is significantly reduced if larger data sets are available.

- SC 6: Eq. 6. I'm a bit baffled by the mix of equation and free text. I have to admit that I'm not sure if that's formally allowed or not. You might think of a variable for "wind-assigned anomaly". Something like "$\delta XCH_4$" or similar. I realize that you would only need this variable at this point so this is only a recommendation. You could also wait for a comment from the type setting of the journal.

Thanks for the suggestion. A variable for the wind-assigned anomaly would be indeed nice, but finding the right name is not so easy, thus we would like to leave it as it is for the time being and wait for comments from the typesetting.

- SC 12: I'm afraid you didn't add your statement to the manuscript. At least in the "track-changed" it is unchanged (see 256 ff). Moreover, you still haven't answered the question. How did you come up with exactly 7 km? Why not 6 or 8 km?

Sorry for this mistake. The statement has been added to the manuscript (Sect. 3.1).

We use 6 km (sorry, 7 km was a typo here) which is adopted from the study by Schneider et al. (2021), where the TROPOMI+IASI $TXCH_4$ products are the partial columns up to 6 km a.s.l. The results derived from the two data sets are then comparable. For TROPOMI+IASI $TXCH_4$ products, ground surface to 6 km a.s.l. has been chosen, because it represents the layer for which the DOFS (degree of freedom of signal, i.e., the sum of the diagonal elements of the averaging kernel) of the combined profile product is generally very close to 1.0, meaning that the combined satellite product is well sensitive to variation of these partial column amounts

- SC 13/14: "This cone plume model only considers a simple linear proportion of wind speed and emission strength. Huge biases are expected in a simple day or in a short period. But these biases can be compensated over a long-term period." If you do not explain how these biases come about, it is hard to understand why they are compensated over a long period.

   I do not understand why you expect such huge biases. Most plume models (e.g. Gaussian plume model) are recursively linear with wind speed and linear with emissions rate. Usually, in plume models some parameter accounts for turbulence. The only possible representation of turbulence in your model is in the angle α=60°. In some cases, this angle will be too small, in some cases it will be too large. This effect might be canceled out, as you suggested, over the long observation period. But it seems as your cone-model is either showing lower $XCH_4$ enhancements, or supposedly too narrow plumes. As your overall goal is to be representing for the overall observation period this is no show-stopper, but please openly discuss/explain the limits of the cone plume model to the reader.
   In your former publication you showed plots of the cone plumes for different values of α. While you derived 60° as the best fit to the $N_2O$ plumes for Madrid, I'm not convinced that 60° is necessarily the optimal fit for the USCB, too. At the very least should investigate to what

quantitative extent variations in α impose uncertainties on your emission estimate. As far as I can see this has not yet been considered in your uncertainty analysis Sect. 3.3.

Thank the referee for pointing it out. The assuming opening cone α can be either too small or too large in different single days. For example, the figures below show enhancements from CAMS $XCH_4$ and the simple cone plume model on January 6, 2018. The angle α is overestimated, which results in lower values compared to the CAMS $XCH_4$ enhancements. This effect can be canceled out over the long-term observations.

[Figure]

The estimated emissions over the study period show a positive correlation to the α values (see figure below, the blue horizontal line represent the total value of the CAMS-GLOG-ANT inventory). The emission rate derived from CAMS $XCH_4$ fits the best to the CAMS-GLOB-ANT inventory for α=60°. This finding supports our empirical choice for α (over a long-term period).

[Figure]

We have added a subsection to the Sect.3.3 and discussed the impact of a suboptimal choice for α. The choice of the cone angle introduces small uncertainties to the averaged results. The subsection below has been added in the manuscript:

**3.3.4 Investigation of different choices for angle of the emission cone**

The angle ($\alpha = 60°$) used in the simple cone plume model is an empirical value which affects the deduced emission strengths. Figure 11 shows the results when $\alpha$ is decreased or increased by 10°. Changes in the spatial distributions of wind-assigned anomalies and in the correlations derived from

CAMS and the simple cone plume model are nearly negligible when using different angles in the model. The estimated emissions are $789 \pm 16$ kt/a ($9.5E26 \pm 1.9E25$ molec./s) for $\alpha = 50°$ and $832 \pm 17$ kt/a ($9.9E26 \pm 2.0E25$ molec./s) for $\alpha = 70°$, which are 3% lower and 2% higher than that with using the empirical angle ($\alpha = 60°$).

[Figure]

**Figure 11: Similar figures to Figure 5b-c. Results are derived from CAMS XCH₄, CAMS emission inventory and ERA5 wind at 330 m for (a)-(b) $\alpha = 50°$, and (c)-(d) $\alpha = 70°$.**

- SC 16 Figure 6: Why do the colorbars start with the value 5 and not 0? If the colorbars are extended (i.e. $\Delta XCH_4 < 5$ is the same color as $\Delta XCH_4 = 5$), an extension arrow at the colorbar should indicate so. The same applies for the upper end of the colorbar. But, as mentioned in the general comment above, I strongly recommend using the same range of values for all colorbars. If the spatial pattern in (c) would not be recognizable anymore you can leave the colorbar as it is now., but at least start your colorbars in (a) and (b) with 0.

  Also indicate the different colorbars in the caption. Currently it seems like the modelled plumes fit perfectly to the CAMS forecast and the TROPOMI data. Which they do only in spatial appearance, not concerning the magnitude of XCH₄ enhancement.

Thanks for this noting. Fig. 6 (a) and (b) have been updated. For Fig. (c), we mentioned that different colorbars are used. Note, the enhancement is always positive based on the simple cone plume model (c) and thus, no extension is applied to the colorbar.

[Figure]

**Figure 6:** ΔXCH₄ together with the ERA5 wind at 12:00 UTC from (a): TROPOMI observations at 11:34 UTC, (b): CAMS forecast at 12:00 UTC, and (c): from the simple cone plume model (averaged over the daytime) based on the CAMS-GLOB-ANT inventory over the USCB region on an example day (6 June 2018). The "bg" in the title of (a) and (b) represents the average background, derived from the mean XCH₄ in the upwind region (50.3º-50.8º N, 19.5º -20.0º E). Note, a different colorbar has been used in panel (c) for improved recognizability.

- SC 17: Sorry, in my first review I mixed the terms „colormap" and „colorbars" by mistake. Here, I was actually referring to the colorbars and the different range of values covered by it. As mentioned in SC 16 this is ok, if you mention in the text why you did so.

Thanks, no problem – please see the reply above.

- SC 20: I actually think that these two plots are of high explanatory value. Please consider including them in the manuscript (optionally). Especially, because I now realized that I misunderstood, that by NE/SW in the title of your figures you mean wind coming from NE/SW and not plumes propagating in NE/SW-direction. My bad, but if you want to make sure that this will not be misunderstood by the reader, you could include these two plots in the manuscript.

Thanks. As the referee suggested, these two figures provide additional supporting information for readers to better understand the applied methods. They have therefore been added in the appendix.

- SC 24/27: In the abstract you give your emission estimate with 479±4 kt/yr and 437±18 kt/year. Then you share the results from your sensitivity analysis regarding wind speed, separately. The uncertainties given your sensitivity analysis should be included in the uncertainties of your emission estimates. The CH₄ exhaust from the ventilation shafts is released at a height of approx. 20 m. Propagating downwind it will be carried upwards by convective eddies and thereby distributed in the entire boundary layer. Have a look into the video supplement of https://amt.copernicus.org/articles/14/2717/2021/#section11 or the Figure A2. While the highest concentrations of CH₄ will, for sure, be advected in the middle of boundary layer, or even closer to the ground, you'll need to include the vertical variations of wind speed in your emission estimate uncertainties in some way!

The referee is correct in assuming that we did not consider the vertical variations of wind speed. Instead, we use wind speed and direction at a certain (hopefully) representative altitude to derive the emissions. For this reason, we discussed the sensitivity resulting from using the wind field at different levels (10 m and 500 m). Please note that other current methods for quantifying emissions also use the simplification of a 2-dimensional transport in the planetary boundary layer, e.g., Liu et al. (2021). We are investigating the possibility of extending our method by incorporating a 3-dimensional plume dispersion model.

(Liu, M., van der A, R., van Weele, M., Eskes, H., Lu, X., Veefkind, P., de Laat, J., Kong, H., Wang, J., Sun, J., Ding, J., Zhao, Y. and Weng, H.: A New Divergence Method to Quantify Methane Emissions Using Observations

of Sentinel-5P TROPOMI, Geophys. Res. Lett., 48(18), e2021GL094151, doi:https://doi.org/10.1029/2021GL094151, 2021.)

> In your uncertainty analysis (Sect. 3.3) you analyze three sources of uncertainty. Within the analysis the emission results vary strongly from the validation emission estimates given in the abstract and conclusion. Considering this, I'm confused how the uncertainty in your overall emission estimate can be so small. Please state the relative contributions of the sources of uncertainty to the overall uncertainty, including uncertainties induced by the selection of cone opening angle α (see SC 13/14)

The previous uncertainties in the emission estimate are determined by considering the deficits of the background model due to the imperfect elimination of the background. The uncertainties related to the noise in the data set (e.g., the noise in the satellite observations) were not included. In the newly revised manuscript, we have updated the uncertainty values to include noise uncertainties based on the error propagation. The sentence concerning the uncertainties of the emission results has been added to Sec. 2.3:

> "The uncertainties (± values) in the emission estimate are determined by considering the deficits of the background model due to the imperfect elimination of the background and the noise in the data set."

Meanwhile, further error sources as discussed in Sect. 3.3 are included in the complete uncertainty budget (as collected in Table A- 2) for the specification of the total uncertainty (in percentage) on the emissions we deduced (as the reader would expect). The following text has been added to Sect. 3.3:

> "The changes in the estimated emission rates for different products due to different error sources are summarized in Table A- 2. Based on the error propagation, the total uncertainty in the estimated emission rates from the different error sources (background removal and noise in the data, vertical wind shear at 500 m, wind field segmentation, and opening angle $\alpha = 70°$) is approximately 14.7% for CAMS XCH$_4$, 14.8% for TROPOMI XCH$_4$ and 11.4% for TROPOMI+IASI TXCH$_4$. Note that, the use of narrowed angular wind regimes is not a preferable way due to few amounts of data in narrowed wind regimes and thus, is not considered an error source. In addition, the 500 m wind shear was used as a contribution to the budget, as the 10 m wind is not expected to be representative of the PBL."

**Table A- 2: Changes in the estimated emission rates for different products when using different input data or under different situations compared to their results using the default setting (wind at 330 m, NE-SW wind segmentation, $\alpha$ =60°, CAMS-GLOB-ANT for CAMS XCH$_4$ and CoMet inventory for TROPOMI XCH$_4$ and TROPOMI+IASI TXCH$_4$).**

|  | CAMS XCH$_4$ | TROPOMI XCH$_4$ | TROPOMI+IASI TXCH$_4$ |
|---|---|---|---|
| Background removal & noise in the data | 2.1% | 3.6% | 6.1% |
| Vertical wind shear (500 m) | 13.4% | 6.8% | 5.8% |
| wind field segmentation (N-S) | -5.2% | 12.7% | 7.7% |
| angle of the emission cone ($\alpha$=70º) | 2.1% | 0.07% | -0.02% |
| **Total:** | **14.7%** | **14.8%** | **11.4%** |

**Technical comments:**

> • TC 50: In the caption it still says "simple plume model" although the authors have confirmed to switch to the term "plume cone model". Please watch out for consistency.

Sorry for this mistake. Changed accordingly.

- TC 51: In Eq. 2 the indices "i" are still not subscripted. Actually, d and $\Delta CH_4$ are functions of the location. So $x_i$, $y_i$ should not be subscripted, only the index "i":

$$\Delta CH_4(x_i, y_i) = \frac{\varepsilon}{v \cdot d(x_i, y_i) \cdot \alpha}$$

Thank you for clarifying it again. Changed accordingly.

- TC 57: caption Fig. 5: "colorbars in (d) and (e) are different from that for XCH₄". Actually, all four colorbars cover different values. To me it makes sense to have different colorbars for XCH₄ and TXCH₄, as TXCH₄ generally shows higher enhancements. But why different colorbars for (a) & (b)? And why different colorbars for (d) & (e)? See general comment.

Please see our reply to the general comment above.

- TC 67: My apologies, by my phrasing "... among each other ..." it was not clear what I actually meant. I thought of something like that:

[Figure]

So that triangles are actually vertically aligned with the squares. In this plot the reader should become aware that CAMS-GLOB-ANT emissions are always higher than from the CoMet inventory. By plotting them vertically aligned this becomes more obvious.

Many thanks for helping with the improvement of the figure. The figure has been updated:

[Figure]

- TC 71: "We use $NE_{1/2}$ for 0°-90°, $SW_{1/2}$ for 180°-270°, $NW_{1/2}$ for 270°-360°, and $SE_{1/2}$ for 90°-180°". My recommendation was to use the subscript "1/2" everywhere when the wind field is divided into two halves (i.e. everywhere before Sect. 3.3.2). The subscript "1/4" was supposed to be used, when the field is divided into quarters (previously designated by "_narrow"). This indexing is of course only necessary when talking about the narrowed angular wind regimes in Sect. 3.3.2. So, if you don't want to include an index in the manuscript before this Sect. it's fine by me. But in Sect. 3.3.2 it should be "1/4".

We use NE (or SW) covering a range of 180° in our pervious analysis. The narrow fields in Sect. 3.3.2 represent half of the predefined NE (or SW) range, and thus, a subscript of "1/2" is used here. This definition seems to bring some misunderstanding. We changed the "1/2" to "1/4" as the referee recommended. The corresponding text and the caption in Figure 9 (a) and (c) have been changed.